# Metabolite annotation from knowns to unknowns through knowledge-guided multi-layer metabolic networking

Zhiwei Zhou [1,4], Mingdu Luo[1,2,4], Haosong Zhang[1,2], Yandong Yin[1], Yuping Cai[1] & Zheng-Jiang Zhu [1,3] ✉

Liquid chromatography - mass spectrometry (LC-MS) based untargeted metabolomics allows to measure both known and unknown metabolites in the metabolome. However, unknown metabolite annotation is a major challenge in untargeted metabolomics. Here, we develop an approach, namely, knowledge-guided multi-layer network (KGMN), to enable global metabolite annotation from knowns to unknowns in untargeted metabolomics. The KGMN approach integrates three-layer networks, including knowledge-based metabolic reaction network, knowledge-guided MS/MS similarity network, and global peak correlation network. To demonstrate the principle, we apply KGMN in an in vitro enzymatic reaction system and different biological samples, with ~100–300 putative unknowns annotated in each data set. Among them, >80% unknown metabolites are corroborated with in silico MS/MS tools. Finally, we validate 5 metabolites that are absent in common MS/MS libraries through repository mining and synthesis of chemical standards. Together, the KGMN approach enables efficient unknown annotations, and substantially advances the discovery of recurrent unknown metabolites for common biological samples from model organisms, towards deciphering dark matter in untargeted metabolomics.

The metabolome refers to the complete collection of small molecules in living organisms[1–4]. It includes not only endogenously produced known metabolites from cellular metabolism, but also unknown metabolites generated from microbiota, plants, foods, and xenobiotics[3,5,6]. Liquid chromatography–mass spectrometry (LC–MS) based untargeted metabolomics allows to measure thousands of metabolic features from biological samples[7,8]. These metabolic features come from known and unknown metabolites, as well as different ion forms generated during ionization, such as adducts, isotopes, neutral losses, and other in-source fragmentation products[9,10]. Metabolite identification remains the central bottleneck in LC–MS-based untargeted metabolomics[4,11]. For annotation of known metabolites, the most commonly used approach is to search the exact mass of

precursor ion (MS1 $m/z$) and tandem mass spectrum (MS2 spectrum) against standard spectral libraries[12,13]. In the past decade, significant efforts have been made to expand the coverage of spectral libraries. For annotation of unknown metabolites, due to a lack of knowledge of chemical structures, additional experiments or in silico tools were usually required[5,11]. For example, Tsugawa and colleagues employed stable-isotope labeling to determine formulas of unknown metabolites by identifying the labeled and non-labeled pair of metabolic peaks[14]. In addition, bioinformatic tools, such as MetFrag[15], CFM-ID[16], MS-FINDER[17], and SIRIUS[18], have been developed to predict in silico MS/MS or molecular fingerprinting to elucidate unknowns. These tools largely rely on existing structural databases (e.g., HMDB[19] and PubChem[20]) to retrieve putative chemical structures. Therefore, it is

[1]Interdisciplinary Research Center on Biology and Chemistry, Shanghai Institute of Organic Chemistry, Chinese Academy of Sciences, Shanghai 200032, China. [2]University of Chinese Academy of Sciences, Beijing 100049, China. [3]Shanghai Key Laboratory of Aging Studies, Shanghai 201210, China. [4]These authors contributed equally: Zhiwei Zhou, Mingdu Luo. ✉e-mail: jiangzhu@sioc.ac.cn

not feasible to identify unknown metabolites absent in the databases. Instead, in silico approaches for generating new metabolite structures (e.g. MINE[21], BioTransformer[22]) are developed to complement the metabolite coverage. For example, COSMIC was recently developed to annotate unknowns from in silico-generated metabolites[23]. More recently, MSNovelist provided a de novo structure generation approach from mass spectra for unknown metabolite annotation without a requirement of structural databases[24].

Alternatively, network-based approaches are increasingly adopted in untargeted metabolomics for metabolite annotation, especially for unknown metabolites without available standard MS/MS spectra[25-28]. The most prominent network approach is molecular networking (MN) in GNPS, which is an MS/MS similarity network that links mass spectra of metabolites based on spectral similarity[28,29]. Various in silico approaches, such as NAP[30], MS2LDA[31], MolNetEnhancer[32], and others[33], were further developed and combined with in silico MS/MS fragmentation tools to infer metabolite structures in GNPS molecular networking. In addition, mass difference annotation[34,35], taxonomic[36], and chemical class[37] information could be added to propagate annotation or re-rank in silico annotated candidates. Recently, ion identity information (e.g., adducts, in-source fragments) can be integrated into molecular networking to cluster different ion species of the same metabolite and remove redundant network connections[38,39]. Unlike molecular networking, peak correlation network approaches employ peak intensity, chromatographic peak shape, mass difference and other information to construct the network, wherein the linked metabolic features are regarded as potential biological associations and beneficial to metabolite annotation[40-46]. For example, NetID used an integer linear programming approach to optimize a peak correlation network[41]. NetID improved the accuracy of peak assignments and provided a possible formula transformation between peaks. In general, these data-driven network approaches build networks from experimental metabolomics data to aid metabolite annotations.

Compared to data-driven network approaches, knowledge-based network approaches provide valuable complements from biochemical knowledge[25,47-49]. For example, iMet combined reactant pairs from KEGG and MS/MS spectral similarity to train a classifier model for metabolite annotation[47]. MetDNA is a typical computational tool to combine knowledge-based networks and MS/MS spectral similarity for metabolite annotation[49]. It used a metabolic reaction network (MRN) to connect metabolic peaks with MS/MS spectral similarity in a recursive manner, achieving high coverage and efficient metabolite annotation[49]. Such a knowledge-guided approach enables to preferentially link metabolic peaks with definitive chemical reaction relationships. However, this approach cannot annotate unknown metabolites which are not covered in the knowledge network. Despite accumulative progress in developing network-based methods for metabolite annotation, most of these studies are primarily limited to one major network embedded with different chemical information. Integration of multiple data-driven and knowledge-based networks for metabolite annotation has been increasingly appreciated but remains unrealized due to the lack of appropriate technologies[25].

In this work, we developed an approach, namely, a knowledge-guided multilayer network (KGMN), to enable global metabolite annotation from knowns to unknowns in untargeted metabolomics (Fig. 1). The KGMN approach integrated three layers of networks, including a knowledge-based metabolic reaction network (KMRN), a knowledge-guided MS2 similarity network, and a global peak correlation network. We first demonstrated that this multilayer network strategy significantly improved the identification accuracy of known metabolites to >95%. Furthermore, we demonstrated the principle of metabolite annotation from knowns to unknowns using KGMN in an in vitro enzymatic reaction system and different biological samples, with ~100–300 putative unknowns being annotated in each data set. Most importantly, more than 80% of unknown metabolites were

corroborated with other in silico MS/MS tools. Finally, we evaluated putative unknown metabolites whether are recurrent in similar samples in the metabolomics repository[50]. We successfully discovered five unknown metabolites that are absent in common MS/MS libraries by integrating KGMN and repository-mining. Altogether, the KGMN approach allows efficient annotations of unknown metabolites and substantially advances the discovery of recurrent unknowns toward deciphering dark matter in untargeted metabolomics.

## Results

### The workflow

Knowledge-guided multilayer network enables global unknown metabolite annotation by propagating annotations from knowns to unknowns along the metabolic reaction network. The KGMN approach integrated three layers of networks: (1) knowledge-based metabolic reaction network (KMRN), (2) knowledge-guided MS2 similarity network, and (3) global peak correlation network. The seed metabolites were first annotated by matching their properties (MS1, retention time, and MS2 spectrum) to metabolite standard libraries (Fig. 1a), and mapped into the metabolic reaction network to retrieve reaction-paired neighbor metabolites (network 1 in Fig. 1b). This network is a knowledge-based metabolic reaction network, where known or unknown metabolites are linked by either known or in silico reactions. Specifically, known metabolites and reactions were retrieved from KEGG, while unknown metabolites and reactions were curated via performing in silico enzymatic reactions using known metabolites in the KEGG database as substrates (see Methods and Supplementary Fig. 1). For example, oxaloacetate (C00036) can be reduced to malate (C00149) with a reductase. Such a reduction reaction could be applied to other structurally similar metabolites and generate possible unknowns with novel structures (Supplementary Fig. 1a). These unknown products are linked with their precursors to expand the metabolic network from known chemical space to unknown space. In sum, a total of 34,858 unknown metabolites were generated from known metabolites, and further linked together through 52,137 edges and 1504 biotransformation types in KMRN (Supplementary Fig. 1b–d). These unknown metabolites included 405 known-unknowns and 34,453 unknown-unknowns, depending on whether they were included by HMDB (version 4.0, released on 2018-12-18).

In KGMN, network 1 provides the known-to-unknown metabolic reaction knowledge to guide the construction of the MS2 similarity network from the LC–MS/MS data (network 2 in Fig. 1b). Specifically, reaction-paired neighbor metabolites (knowns or unknowns) from seeds were retrieved from network 1. Their calculated MS1 *m/z* and predicted RTs were matched to the experimental values in the data file. Meanwhile, the surrogated MS2 spectra from seed metabolites were also used for MS/MS spectral match. The matched peaks were annotated as putative neighbor metabolites and linked to seeds in network 2. Thereby, seeds are linked to other annotated metabolites with four constrains, including MS1 *m/z*, RT, MS/MS similarity, and metabolic biotransformation (e.g., reduction, +2H; decarboxylation, -CO$_2$). Although we named network 2 as the knowledge-guided MS2 similarity network, in fact, the connections between nodes in the network had four constrains (see Methods). Compared to GNPS and other tools which solely use the MS/MS similarity to construct the network, our knowledge-guided MS2 similarity network has explicable structural relationships between two nodes (i.e., metabolic reaction biotransformation) and a more succinct network topology (Supplementary Fig. 2). In addition, similar to MetDNA, the annotated metabolites could be used as new seeds to annotate more metabolites and extend the network. Such annotation is performed in a recursive manner until there is no new metabolites annotated in LC–MS/MS data.

The global peak correlation network aims to annotate different ion forms in LC–MS data (network 3 in Fig. 1b). This network utilizes chromatography co-elution correlation to recognize different ion

 

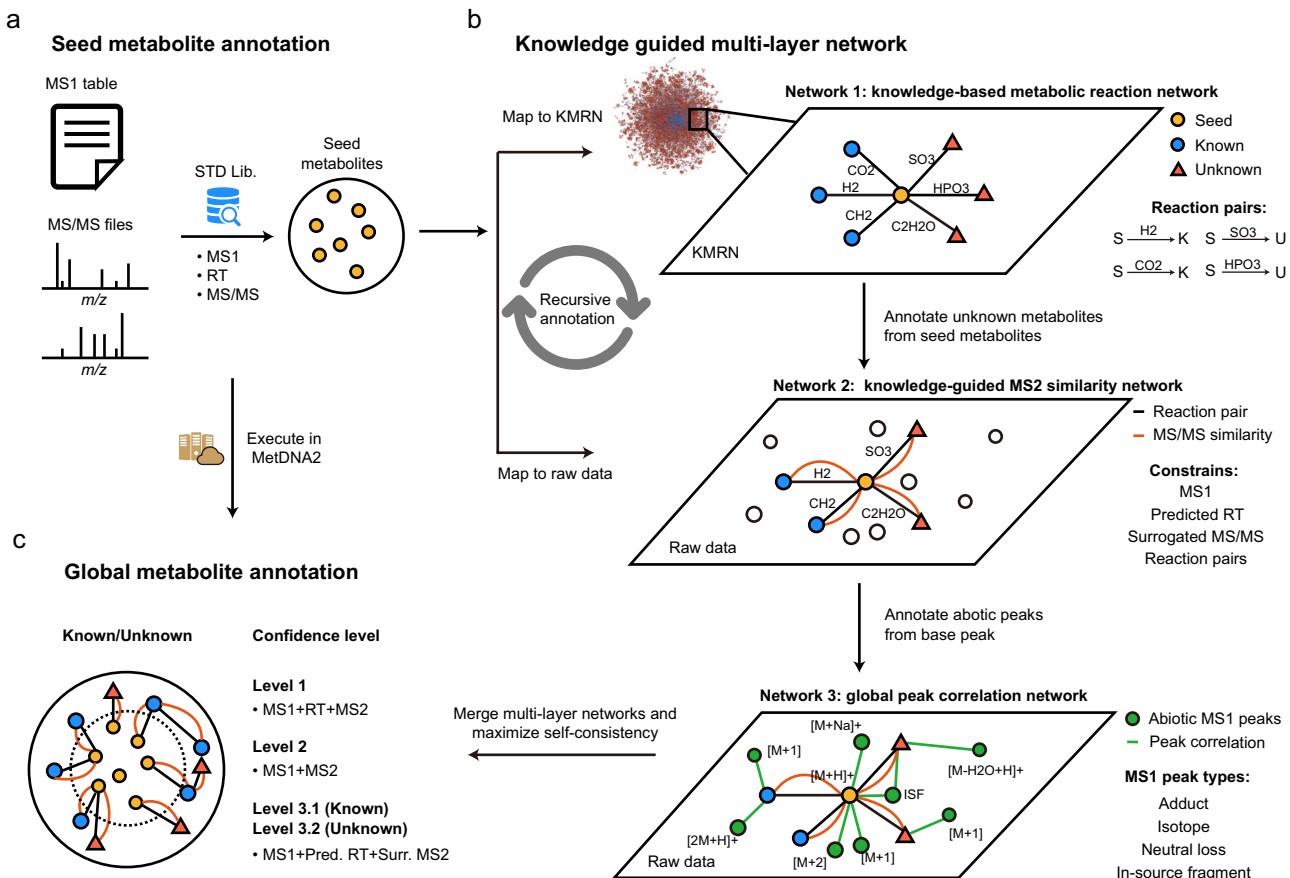

**Fig. 1 | The workflow of knowledge-guided multilayer network (KGMN) for metabolite annotations in untargeted metabolomics. a** Annotation of seed metabolites by matching with the standard library using MS1, retention time (RT), and MS/MS spectra. **b** Knowledge-guided multilayer network enables global metabolite annotations by propagation from knowns to unknowns. Network 1 is a knowledge-based metabolic reaction network (KMRN), where known or unknown metabolites are linked by known and in silico metabolic reactions. Network 2 is a knowledge-guided MS2 similarity network, where annotations propagate from known to unknown along KMRN. Network 3 is a global peak correlation network to recognize different ion forms and improve the accuracy of peak assignment. **c** The KGMN-based metabolite annotations are executed in an automated and unsupervised manner and reported with definitive confidence levels.

forms, including adducts, isotopes, neutral losses, and in-source fragments (ISF). Briefly, all annotated peaks from the knowledge-guided MS2 similarity network were considered as base peaks. Next, different ion forms derived from each base peak were extracted from the peak list in LC–MS data. More specifically, common adducts (e.g., Na⁺, K⁺) and neutral losses (e.g., -H₂O, -NH₃) were searched within co-eluted peaks, while in-source fragments were retrieved from the MS2 spectra of the base peaks (See Methods and Supplementary Fig. 3). Then, a peak correlation subnetwork is constructed for each annotated metabolite through connecting base peak and different ion form peaks. The subnetwork describes the comprehensive peak profiles of the metabolite during ionization in mass spectrometry measurements. As a result, a global peak correlation network (network 3) is constructed by combining subnetworks. Similar to NetID, the global peak correlation network provides a valuable basis to optimize and filter metabolite annotation from the first two-layer network and improve the accuracy of peak assignment (Supplementary Fig. 4). Peak annotations are compared and scored within and across subnetworks, while conflicts are further resolved by maximizing the self-consistency in each subnetwork. The subnetworks with the most linked edges are reserved, while unsatisfactory and conflict subnetworks and their putative annotations are removed. Most importantly, unlike other peak correlation network approaches such as CAMERA[44] and IIMN[38], our global peak correlation network can effectively filter putative metabolite annotations from the first two-layer networks.

Finally, the annotation results from KGMN are given with definitive confidence levels according to the MSI guidelines[51] (Fig. 1c). The KGMN approach is implemented and freely available in the MetDNA2 webserver (http://metdna.zhulab.cn/). It supports multiple metabolomics workflows (Supplementary Table 1), and accepts various data imports from common data processing tools, including XCMS[52], MS-DIAL[53], and MZmine[54]. The network visualization and interactive investigation is performed via Cytoscape. We have provided a series of tutorials to visualize the network and connect KGMN result with in silico MS/MS workflow and repository search in Supplementary Notes 1–3.

## KGMN improves peak annotation accuracy

The KGMN approach enables to optimize and filter metabolite annotation and improves the accuracy of peak assignment through a global peak correlation network (see Methods and Supplementary Figs. 3, 4). Here, we demonstrated the principle with an example in an NIST urine sample (Fig. 2a). Metabolic features M285T555 and M153T555 were putatively annotated as xanthosine and 5-ureido-4-imidazole carboxylate, respectively. Base peaks of both metabolites and their related ion forms were extracted to construct subnetworks. Two subnetworks had 14 and seven recognized peaks, respectively. One conflict peak assignment was observed in these subnetworks. The base feature of M153T555 was assigned as an in-source fragment ion of xanthosine. To resolve this conflict and maximize the self-consistency of peak annotations in two subnetworks, the subnetwork of M153T555 and its

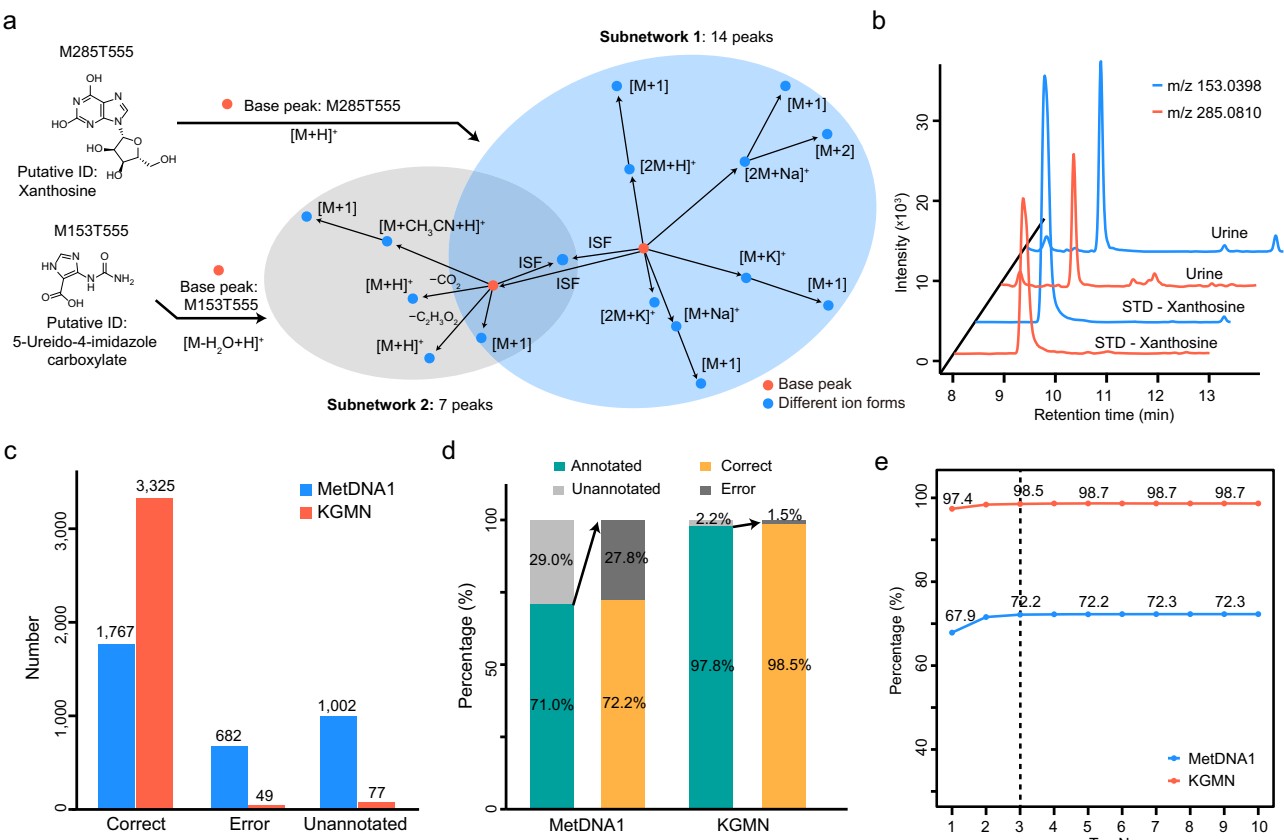

**Fig. 2 | KGMN improves peak annotation accuracy. a** Peak correlation subnetworks for metabolic features of M285T555 and M153T555. The M153T555 was correctly recognized as the in-source fragment of M285T555 by KGMN. The feature identifier is defined as the (nominal) mass and RT in seconds (e.g., M153T555: nominal mass 153, retention time 555 s). **b** The validation of M153T555 (m/z 153.0398) as an in-source fragment of M285T555 (m/z 285.0810) using the chemical standard of xanthosine. **c–e** The improved accuracy of known metabolite annotation using MetDNA1 and KGMN: c, statistics of annotation numbers; d, percentages of annotation coverage and correct/error rates; **e** correct and error rates among top *n* annotations. The x-axis represents the top n metabolite candidates for each peak. Source data are provided as a Source Data file.

annotation of 5-ureido-4-imidazole were removed. As a result, all peak annotations in two subnetworks could be merged and confirmed to be associated with xanthosine. We further validated that M153T555 was an in-source fragment of xanthosine using the chemical standard (Fig. 2b).

Then, we systemically evaluated the improved accuracy of peak annotations with a manually curated data set which contains a total of 242 metabolites, 3451 metabolic features from five different biological samples, and two ionization modes (Supplementary Table 2 and Supplementary Data 1). These metabolites were first identified in each biological sample by MS1 match, RT match, and MS/MS match with standard libraries. Then, metabolite identifications and related ion form annotations were manually checked and labeled for accuracy evaluation (see Methods and Supplementary Fig. 5). Among these metabolic features, our previously developed MetDNA (denoted as MetDNA1) reported a total of 2449 annotations, including 1767 correct (72.2%) and 682 error (27.8%) annotations. The remaining 1002 peaks were not annotated (Fig. 2c). As a comparison, with the optimization and filtering of the global peak correlation network, the KGMN approach significantly increased the correct peak annotations to 3325 (98.5%) and decreased error annotations to 49 (1.5%; Fig. 2c). As shown in Fig. 2d, when only the annotated peaks were considered, annotation coverage increased from 71.0 to 97.8%, and correct peak annotations increased from 72.2 to 98.5%. Considering different metabolite annotations for one peak, correct annotation rates were also consistently improved (Fig. 2e). Similar results were also obtained for individual data sets in both positive and negative ionization modes

(Supplementary Fig. 6). We also compared the global peak correlation network in KGMN with CAMERA[44]. Compared to CAMERA, KGMN annotated more ion forms (3374 vs 2297, KGMN vs CAMERA), and had a higher correct rate (97.5 vs 81.7%, KGMN vs CAMERA) (Supplementary Fig. 7). In particular, KGMN has excellent performances in recognizing inexplicable in-source fragmentation ions and neutral losses (Supplementary Fig. 7b). Overall, these results demonstrated that the KGMN approach effectively extended annotation coverage and increased annotation accuracy for untargeted metabolomics.

A salient feature of KGMN is that peak assignment evaluation and optimization are performed for all peaks in automated and unsupervised manners. In addition, we found that this approach was highly effective to recognize false positive annotations caused by in-source fragments. For example, in-source fragment M112T282 was annotated as metabolite cytosine in MetDNA1 because of its high MS/MS match score (0.9817) (Supplementary Fig. 8). This feature was successfully recognized as an in-source fragment feature of metabolite *N*4-acetylcytidine by KGMN approach. More examples are provided in Supplementary Fig. 9. Taken together, these results validate that our KGMN approach provides substantial improvements of peak assignment accuracy, which facilitates accurate annotations of unknown metabolites in complex biological samples.

## Metabolite annotation from knowns to unknowns

To demonstrate the principle of metabolite annotation from knowns to unknowns in KGMN, we experimentally incubated a mixture of 46 common metabolites (46std_mix) with a human liver

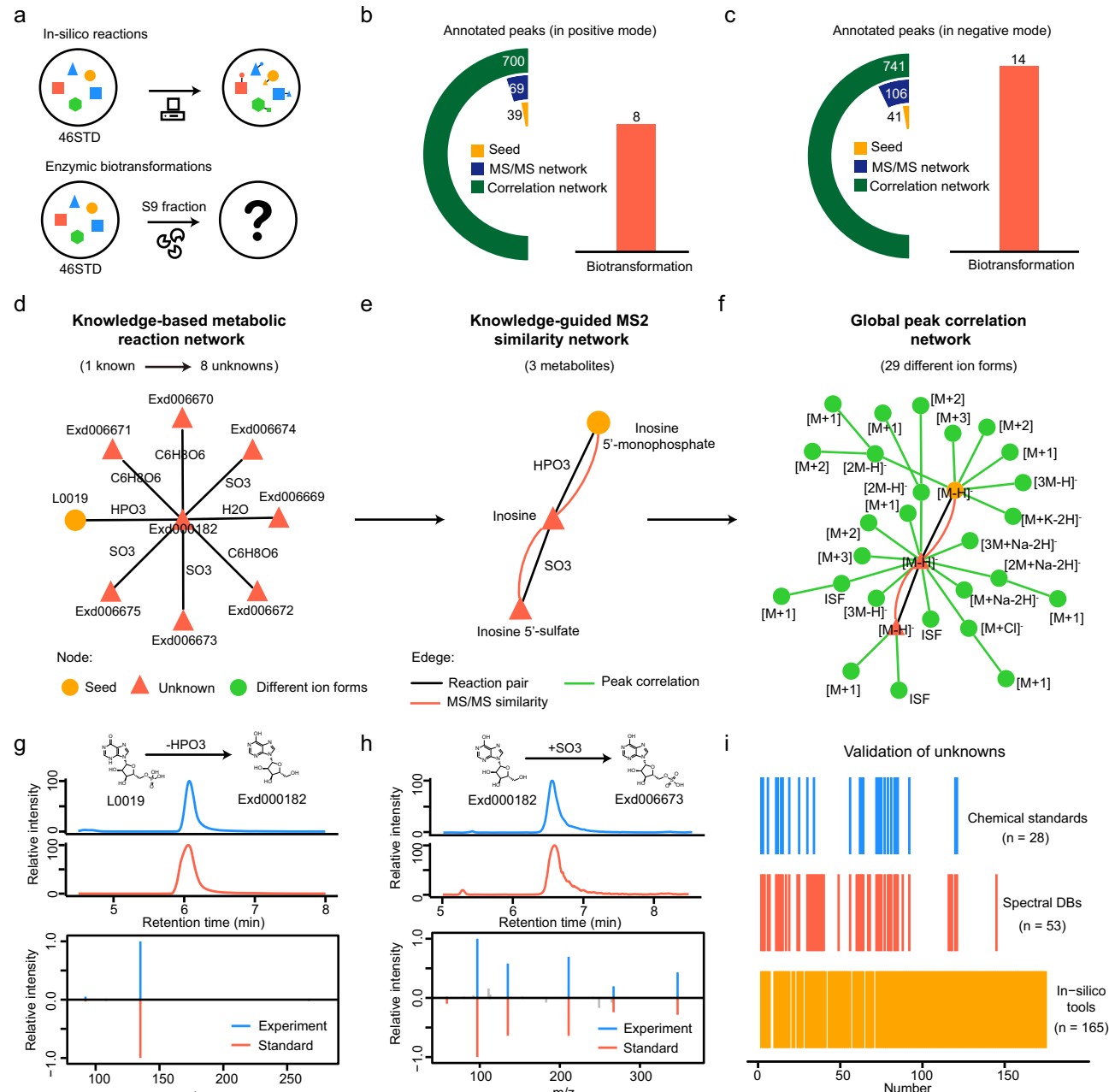

**Fig. 3 | Metabolite annotation from knowns to unknowns. a** Generation of unknown metabolites from a mixture of 46 metabolites (46std_mix) with in silico reactions or enzymatic biotransformation via human liver S9 fraction incubation. **b**, **c** Annotated peaks in positive (**b**) and negative ionization modes (**c**); the left cyclic bars represent the annotated peaks in different networks. The right bar represents the involved biotransformation in an unknown annotation. **d**–**f** Unknown annotations from seed metabolite inosine 5′-monophosphate (IMP, L0019): **d** IMP generates eight unknowns through four transformations in the knowledge-based metabolic reaction network; **e** knowledge-guided MS2 similarity network annotates two unknowns from the seed; **f** 29 different ion forms were annotated from three metabolites in global peak correlation network. **g**, **h** Validation of two annotated unknowns using chemical standards: inosine (**g**, labeled as Exd000182) and inosine 5′-sulfate (**h**, labeled as Exd006673). **i** Validation of annotated unknowns with different strategies. Source data are provided as a Source Data file.

S9 fraction for 24 h (Fig. 3a). The liver S9 fraction contains most phase I and phase II metabolic enzymes and has been widely used to investigate in vitro metabolism. Here, we treat the 46 compounds as known seed metabolites, while their in vitro metabolic products are defined as unknowns. Unknown metabolites in the incubation solution were analyzed by LC–MS/MS. To identify unknowns, we constructed the knowledge-based metabolic reaction network from 46 metabolites, including 531 possible unknown structures and 642 reaction pairs (Supplementary Fig. 10 and Supplementary Data 2). This knowledge-based metabolic reaction network was used to

annotate unknown metabolites in LC–MS/MS data. In positive ionization mode, the KGMN approach annotated 39 known and 69 unknown peaks, and a total of 700 MS1 peaks associated with known and unknown metabolites were discovered in the global peak correlation network (Fig. 3b, Supplementary Fig. 10 and Supplementary Data 3). Unknown metabolites were generated from eight types of biotransformation (Supplementary Table 3). Similar results were obtained in negative ionization mode (Fig. 3c). KGMN approach annotated 41 known and 106 unknown peaks, and a total of 741 peaks in the global peak correlation network. Unknown

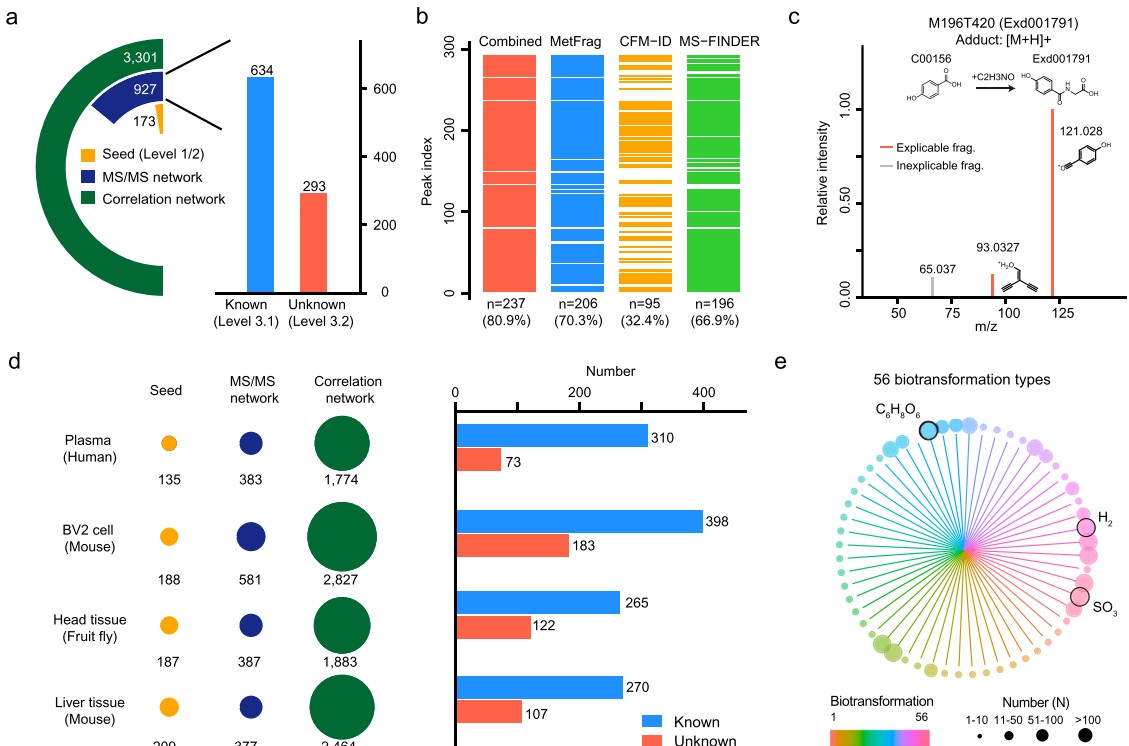

**Fig. 4 | Global annotation of unknown metabolites in biological samples.**
**a** Annotated known and unknown metabolites in NIST human urine samples (positive ionization mode). The left panel is the statistics of annotated peaks in the multilayer networks, and the right panel is the statistics of annotated known and unknown peaks. **b** Corroborations of annotated unknown metabolites with different in silico MS/MS tools. **c** A corroboration example of unknown metabolite 4-hydroxyhippuric acid using in silico MS/MS tools. **d** Global annotations of
metabolites in different biological samples in positive ionization mode. The left panel is the statistics of annotated peaks in the multilayer network, and the right panel is the statistics of known and unknown metabolites. **e** Summary of biotransformation types of annotated unknown metabolites. The color represents different biotransformation types, and the node size represents the frequency number. Source data are provided as a Source Data file.

metabolites were generated from 14 types of biotransformation (Supplementary Table 3).

We further demonstrated the known-to-unknown annotation with an example of inosine 5′-monophosphate (IMP, denoted as L0019 in Fig. 3d–f). As one of the seed metabolites, IMP was identified by matching the standard MS/MS spectral library. Then, the IMP-related-subnetwork was retrieved from the knowledge-based metabolic reaction network (Fig. 3d). Specifically, 8 metabolites were in silico generated from IMP through one- or two-step reactions with four different types of biotransformation, including dephosphorylation (-HPO₃), sulfation (+SO₃), glucuronidation (+C₆H₈O₆) and hydrolysis (-H₂O). With the guidance of the IMP-reaction network, the knowledge-guided MS2 similarity network was constructed (Fig. 3e). In this network, three metabolites were connected via reaction pair (black line) and MS/MS similarity (orange line), including one seed metabolite (IMP) and two unknowns (inosine and inosine 5′-sulfate). To clarify, inosine and inosine 5′-sulfate were specifically defined as unknowns only in this in vitro experiment because they were not included in 46std_mix and were generated through enzymatic reactions. Furthermore, a peak correlation network was constructed with 29 different ion form peaks from three annotated metabolites (Fig. 3f). In this example, inosine was the product of dephosphorylation of IMP, and further was converted to inosine 5′-sulfate through sulfation. We further confirmed identifications of inosine and inosine 5′-sulfate with chemical standards (Fig. 3g, h).

Finally, we validated the accuracy of annotated unknown metabolites with multiple strategies, including chemical standards, public spectral libraries, and in silico MS/MS tools (Fig. 3i and Supplementary Data 3). For all 175 annotated unknowns in positive and negative ionization modes (Fig. 3b, c), 28 (16%), 53 (30%), and 165 (94%) were

validated by chemical standards, public spectral libraries, and in silico MS/MS tools, respectively. Examples in detail are provided in Supplementary Fig. 11. Taken together, these results demonstrated that the KGMN strategy effectively annotates unknown metabolites from knowns and provides reliable putative structures for unknown peaks on a large scale.

## Global annotation of unknown metabolites

To determine the performance for real biological samples, we applied KGMN workflow to the untargeted metabolomics data of NIST human urine samples. In positive ionization mode, 173 seed metabolites were first annotated by matching with the standard library (Fig. 4a). Then, a total of 927 peaks, including 634 knowns and 293 unknowns were annotated through the knowledge-guided MS/MS similarity network (Supplementary Fig. 12). Finally, 3301 MS1 peaks associated with metabolites were annotated in the global peak correlation network (Supplementary Fig. 13). Confidence levels of putatively annotated knowns and unknowns through our KGMN approach were assigned as levels 3.1 and 3.2, respectively (see Methods and Fig. 4a). Similarly, in negative ionization mode, 161 seed metabolites were first annotated and 1,283 peaks including 652 knowns and 631 unknowns were further annotated through the knowledge-guided MS/MS similarity network (Supplementary Fig. 14). These results demonstrated that the KGMN approach significantly expanded the metabolite annotation coverage from seed metabolites in biological samples. In this work, we are interested in these putatively annotated unknown metabolites. Since unknowns are not included in spectral databases and without available chemical standards, we employed common in silico MS/MS tools to corroborate their reliability (Supplementary Data 4). 206, 95, and 196 unknown peaks in positive ionization mode, and 540, 309, 299

unknown peaks in negative ionization mode were corroborated by MetFrag[15], CFM-ID[16], and MS-FINDER[17], respectively (Fig. 4b and Supplementary Fig. 14). In sum, 237 (80.9%) and 547 (86.7%) unknown peaks were corroborated by at least one in silico MS/MS tool in positive and negative ionization modes, respectively. For example, the glycine-conjugated metabolite, 4-hydroxyhippuric acid was corroborated by in silico MS/MS tools (Fig. 4c). More examples are provided in Supplementary Fig. 14.

Finally, we applied KGMN to different biological samples, including human plasma, BV2 cells, fruit fly head tissue, and mouse liver tissue samples. Consistently, about 100–200 (154 ± 36, Mean ± S.D.), 300–600 (607 ± 287), and 2000–3000 (2445 ± 758) peaks were annotated in seed annotation, knowledge-guided MS/MS similarity network, and global peak correlation network in positive ionization mode, respectively (Fig. 4d, Supplementary Table 4, and Supplementary Data 5). Similar results were obtained in negative ionization mode (Supplementary Fig. 14). On average, 100–300 unknown metabolites were annotated in each data set. These metabolites were generated through 56 types of biotransformation (Fig. 4e and Supplementary Table 5). The most frequent biotransformation types included glucuronidation ($C_6H_8O_6$), sulfation ($SO_3$), and oxidation/reduction ($H_2$). Overall, these results demonstrated that the KGMN approach enables global and efficient annotation of unknown metabolites in different biological samples.

### Validation of recurrent unknowns through repository-mining

With global annotation of unknown metabolites, it is feasible to evaluate the recurrence of unknowns in the public metabolomics data repository. Here, we searched our putatively annotated unknowns in NIST human urine samples against GNPS/MassIVE database through MASST[50] (Fig. 5a). A total of 187 unknowns were recurrent in 351 data sets and 13,418 data files (Fig. 5b and Supplementary Data 6). Specifically, 69, 73, 20, and 25 unknowns were detected in 1, 2–5, 6–10, and >10 data sets, respectively. Among them, 76, 25, 13, and 73 unknowns were present in 1–10, 11–30, 31–50, and >50 data files, respectively. These data sets were acquired from 10 different species and 12 different sample types (Fig. 5c). We noticed that recurrent unknown metabolites mainly appeared in human species, such as plasma, serum, and urine, which were the same as the sample types tested in this study.

We further demonstrated an example for a recurrent unknown peak of M262T526, which was observed in seven data sets and 41 data files (Fig. 5d). Through mining their related meta information in GNPS/MassIVE, we found it was reported in multiple species as an unknown, including human (63%), mouse (10%), and plants (27%, e.g., *Solanum lycopersicum*). Interestingly, it was only observed in body fluid (plasma, serum, and urine) instead of tissues in mammals, which indicates that this unknown may come from microbiota or xenobiotic resources (e.g., foods). Our KGMN approach putatively annotated this feature as *O*-sulfotyrosine (Fig. 5e). In the knowledge-based metabolic reaction network, this metabolite can be converted from two possible routes, including sulfation (+$SO_3$) of tyrosine and demethylation (-$CH_2$) of L-tyrosine methyl ester 4-sulfate. In the metabolomics data, tyrosine was first annotated in seed annotation, and its annotation further propagated to *O*-sulfotyrosine with the guidance of a knowledge-based metabolic reaction network. Finally, *O*-sulfotyrosine was annotated with six related ion form peaks in its subnetwork through the global peak correlation network. To confirm this propagated metabolite annotation, we chemically synthesized *O*-sulfotyrosine. The synthetic O-sulfotyrosine showed good consistencies in both retention time and MS/MS spectrum with the unknown peak in a human urine sample (Fig. 5f). At the time of writing our study, this metabolite is not included in common metabolite databases such as KEGG, HMDB (v4.0), MoNA, and GNPS. According to the definitions in the previous literatures[5,55,56], this metabolite is defined as a known-unknown

metabolite. O-sulfated metabolites are products of the co-metabolism of microbes and their hosts and function as a class of key regulators for interaction between microbes and their hosts. This characteristic is in concordance with its recurrent distributions in various sample types (Fig. 5d). Another example of unknown peak M196T420 was observed only in human and mouse samples (Fig. 5g). It was annotated as a known-unknown metabolite of 4-hydroxyhippuric acid from the seed metabolite 4-hydroxyanilian in KGMN (Fig. 5h). Similarly, this annotation was validated using the chemical standard (Fig. 5i). We also validated another three propagated metabolites in Supplementary Fig. 15. Taken together, we demonstrated that the combination of KGMN and repository-mining facilitated validations of recurrent unknowns and advanced understanding of potential origins and biological functions of the newly discovered unknown metabolites. These recurrent metabolites expanded the coverage of potential bioactive metabolites, and offered deeper biological insights of physiological and pathology mechanisms, like microbiota-metabolite-host interactions.

### Discussion

Despite accumulative progress in developing network-based methods for metabolite annotation, most of these studies are primarily limited to one major network imbedded with different chemical information. Integration of multiple data-driven and knowledge-based networks for metabolite annotation has been increasingly appreciated but remains unrealized due to the lack of appropriate technologies. In this work, we developed the knowledge-guided multilayer network approach to enable global metabolite annotation from knowns to unknowns in untargeted metabolomics. We demonstrated that the KGMN approach substantially advanced the discovery of unknown metabolites toward deciphering dark matters in biological samples. Compared to previous network-based approaches, the key advancements of KGMN come from two aspects, including the utilization of metabolic reaction knowledge and proper integration of multilayer networks. The first characteristic of KGMN is using known metabolic reactions to in silico curate unknown metabolites. These unknown metabolites are further linked with their precursors to expand the metabolic network from known chemical space to unknown space. The generated knowledge-based metabolic reaction network (network 1) provides known-to-unknown biotransformation knowledge to guide the construction of the MS2 similarity network from LC–MS/MS data (network 2). Although a few studies have also reported the use of in silico reactions to annotate unknowns in biological samples, these curated unknowns were simply used as an alternative compound database[23,57]. As a comparison, KGMN uniquely uses the knowledge of reaction relationships to expand classic metabolic reaction networks with curated unknown metabolites and provides essential routines for annotation from knowns to unknowns. Such a knowledge-based reaction network provides straightforward interpretations of unknown annotations and effectively reduces the complexity of the MS/MS similarity network generated from untargeted metabolomics data (Supplementary Fig. 2).

The second characteristic of KGMN is the proper integration of three layers of networks as an automated and unsupervised workflow to enable global unknown annotation and improve annotation accuracy. Such an approach provides several strengths, including but not limited to: (1) the well-defined known-to-unknown routines in knowledge-based metabolic reaction network preferentially link metabolic features with definitive chemical reaction relationships; (2) connections between nodes in the knowledge-guided MS2 similarity network have four constrains (MS1 *m/z*, RT, MS/MS similarity, and metabolic biotransformation), which generates explicable structural relationships between two nodes and a more succinct network topology. However, this approach also removes some spectrally similar but hard-to-interpret connections, which may aid to provide interpretations in data-driven networks (i.e., molecular networking). (3) the addition of a global peak correlation network

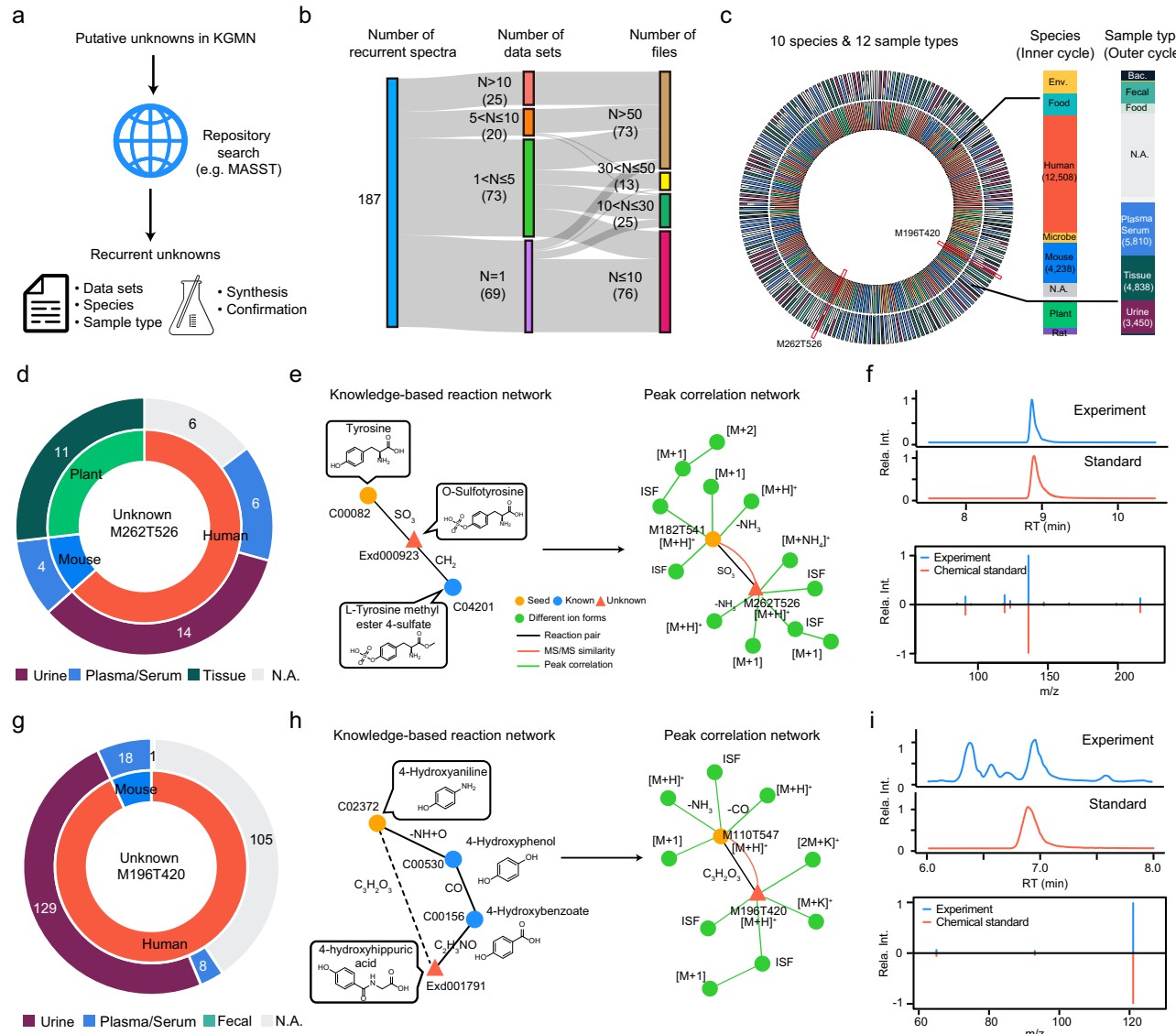

**Fig. 5 | Validation of recurrent unknowns through repository-mining.**
**a** Workflow for validation of recurrent unknown metabolites using repository-mining. **b** Repository-mining statistics of data sets and data files for 186 recurrent unknown metabolites in human urine samples. **c** Repository-mining distributions of 186 recurrent unknown metabolites in species and sample types (left panel). The right panel is the summed distributions of 186 recurrent unknowns in species and sample types. **d–f** Repository-mining and structural validation of a recurrent unknown metabolite (M262T526): **d** recurrent distributions of species and sample types; the inner and outer pie plots are the distributions in species and sample types, respectively; **e** peak of M262T526 was annotated as *O*-sulfotyrosine by KGMN; **f** structural validation using the synthetic standard by matching retention time and MS/MS spectrum. **g–i** Repository-mining and structural validation of a recurrent unknown metabolite (M196T420): **g** recurrent distributions of species and sample types; **h** peak of M196T420 was annotated as 4-hydroxyhippuic acid by KGMN; **i** structural validation using the synthetic standard by matching retention time and MS/MS spectrum. Source data are provided as a Source Data file.

recognizes all possible ion forms, which further increases the coverage of recognized peak number by ~4–5-folds and effectively optimizes and filters putative metabolite annotations from the first two-layer networks. As a comparison, in GNPS, the MS2 spectral similarity network is the main network. To achieve global metabolite annotation, additional bioinformatic tools, such as MetFrag and MS2LDA, or manual interpretation are usually required, and metabolite annotation is performed in multiple steps. Similarly, in NetID, the peak correlation network is the main network. Manual interpretation is usually required to elucidate structures of unknown metabolites in NetID. Although additional chemical information could be added to the main network to further improve the accuracy of metabolite annotation (e.g., ion identity and taxonomic information for GNPS network; MS2 information for NetID), these studies mainly focus on one major network. Therefore, the integration of multiple networks, in particular the

knowledge-based network for metabolite annotation, has been increasingly recognized as a promising strategy. We believe that KGMN is a unique network-based approach that successfully integrates three layers of knowledge-based and data-driven networks for metabolite annotation.

The definition of a multilayer network caused one of the reviewers' discussions during revision. According to a recent review[58], this term lacks a terminology convention, where the authors summarized two types of multilayer networks, "node-colored graphs" and "edge-colored graphs". KGMN is an approach, which integrates the above two types of multilayer networks. Specifically, the relationship between knowledge-guided reaction networks and experimental networks (MS2 similarity network and global peak correlation network) belongs to node-colored graphs, where nodes are not directly shared. The node represents a metabolite in a knowledge-guided reaction network,

while the node represents an experimental peak (or feature) in the MS2 similarity network and the global peak correlation network. In addition, the relationship between the MS2 similarity network and the global peak correlation network belongs to the edge-colored graph. They share nodes but have different edge types. Edge in the MS2 similarity network represents MS2 similarity, while the edge in the global peak correlation network represents ion form relationship. A recent review also termed a similar concept which integrates knowledge and experimental networks as a multilayer network approach[25].

The accuracy of unknown annotation in KGMN depends on several key factors. First, the characterization of MS/MS spectral similarity has a significant impact on the accuracy of unknown annotations. Specifically, although ~60% of reaction-paired metabolites (either known or predicted pair) have MS/MS similarity scores larger than 0.5 (dot product score), the remaining ~40% of reaction pairs have low MS/MS similarity scores even they have high structural similarity (Supplementary Fig. 16). Recently, a similar observation was also reported by the Van der Hooft Computational Metabolomics Group[59]. We believe the implementation of newly developed scoring approaches for MS/MS similarity, like CSS score[60], Spec2Vec[59], and spectral entropy score[61], would further enhance the performance of KGMN. Second, we need to be aware of the challenge to discriminate structurally similar isomers of unknown metabolites. Specific to KGMN, one known metabolite may generate several possible unknown isomers through one biotransformation when performing in silico reaction. For example, isocitrate has four reactive functional groups for glucuronidation, whose product isomers cannot be distinguished effectively in KGMN. To address these challenges, incorporating more orthogonal properties, such as collision cross-section (CCS)[62], would be valuable in the future. Third, we think the accuracy of chemical structures of unknowns is largely dependent on the in silico biotransformation algorithms. In KGMN, we used the predefined reaction sets in BioTransformer, which mainly originated from metabolic reactions in humans. With continuing innovations of in silico reaction tools (e.g., ATLAS[63], CyProduct[64], and MINE2[65]) and the inclusion of metabolic reactions in other species, structural reliability, and coverage of predicted unknown metabolites will be improved, thereby benefiting the performance of the KGMN approach. Finally, although current KGMN has been tested in several common biological samples, one needs to be cautious when applying it to some cases like non-model organisms and environmental and exposomics-related profiles (e.g., waste waters, non-model plants, fungi, and bacteria). It is feasible to incorporate databases for these organisms in the future, like Nature Products Atlas[66] (for bacterial and fungal) and T3DB[67] (for toxin). Through expanding the knowledge of structures and pathways with wider coverages, it is foreseeable that the next versions of KGMN may encompass wider applicability.

Since unknown metabolites are not included in spectral databases and have no chemical standards, validation of unknown annotations remains a grand challenge. In this work, we demonstrated the principle of metabolite annotation from knowns to unknowns in an in vitro enzymatic reaction system and validated this principle with metabolite standards. For unknown annotation in biological samples, we also employed common in silico bioinformatics tools to corroborate structural reliability on a large scale. Even so, it is noteworthy that the confidence levels of all annotated metabolites in KGMN are assigned as level 3 according to the definition of MSI[51]. Metabolite identification still needs to be validated using synthetic chemical standards. In our work, we demonstrated that the combination of KGMN and repository-mining facilitated validations of recurrent unknowns at the repository level. We used such an approach to discover five recurrent unknown metabolites, and further confirmed their annotations with synthetic chemical standards. With the accumulation of open-source data sets in the metabolomics repository, we believe that the KGMN approach will gear to validate more unknown structures through repository-mining.

## Methods

### Chemicals

Pooled human liver S9 fraction (H0610.S9) and NADPH regenerating system (K5100-5) were purchased from Sekisui Xenotech (Kansas City, KS, USA). The cofactors adenosine 3′-phosphate 5′-phosphosulfate lithium salt hydrate (PAPS), acetyl-coenzyme A (acetyl-CoA), uridine diphosphate glucuronic acid (UDPGA) were purchased from Sigma-Aldrich (St. Louis, MO, USA). The glutathione (GSH) was purchased from J&K (Shanghai, China). The NIST urine (SRM 3667) and NIST plasma (SRM 1950) samples were purchased from Ango Biotechnology Co. (Shanghai, China). LC–MS grade methanol (MeOH) and water ($H_2O$) were purchased from Honeywell (Muskegon, MI, USA). LC–MS grade acetonitrile (ACN) was purchased from Merck (Darmstadt, Germany). Ammonium hydroxide ($NH_4OH$) and ammonium acetate ($NH_4OAc$) were purchased from Sigma (St. Louis, MO, USA). Other chemical standards were purchased from Sigma-Aldrich (St. Louis, MO), J&K (Shanghai, China), and TopScience (Shanghai, China).

### Standard MS/MS and RT libraries

In-house MS/MS and RT libraries from chemical standards were used for seed metabolite annotation in KGMN. It supports different types of high-resolution mass spectrometers from various vendors (including Sciex, Agilent, Waters, Bruker and Thermo Fisher). The curation of MS/MS spectral library curation followed the previous publication[68,69]. Briefly, a total of 868 metabolites and 611 metabolites were acquired with 14 collision energies from Sciex TripleTOF 5600/6600 and Thermo Fisher Exploris 480, respectively. Their retention time were also acquired under the Waters BEH Amide column (HILIC) and Phenomenex C18 column (reverse phase). The LC details are provided in LC–MS/MS part.

### Knowledge-based metabolic reaction network

Knowledge-based metabolic reaction network (KMRN) is a network containing known and unknown metabolites (nodes), and their reaction relationship (edges) from known reactions or in silico reactions. The known metabolites and their metabolic reactions were directly downloaded from the KEGG reaction pair database (KEGG RCLASS)[70] on 7 March 2017. It contains 6397 known metabolites and 8129 known reaction pairs, which was described in our previous MetDNA publication[49]. Unknown metabolites were curated from in silico enzymatic reactions with 6397 known KEGG metabolites (Supplementary Fig. 1). The unknown metabolite is defined as the in silico curated metabolites not included in the KEGG database. The BioTransformer[22] (version 1.0.8) was used for in silico enzymatic reactions, and "EC-based transformation" was used for two-step reactions. All curated metabolites were merged with the first layer of InChIKey (14 characters) to remove the stereoisomers. The chemical elements of unknown metabolites were restricted within "CHONPS". As a result, a total of 50,471 unknown metabolites were curated via 193 chemical reactions and 114 enzymes. These curated unknown metabolites had a higher natural product likeness[71] than those in PubChem (Supplementary Fig. 16). The unknown metabolites were further paired with its reactant in an in silico enzymatic reaction, and Tanimoto structural similarity between reaction-paired metabolites was calculated. The reaction pairs with Tanimoto structural similarity larger than 0.7 were reserved. The unpaired metabolites are discarded. The Tanimoto structural similarity was calculated based on PubChem molecular fingerprinting via the R package rcdk (version 3.4.7.1). Finally, both known and unknown reaction pairs were integrated to curate the knowledge-based metabolic reaction network, including 41,336 nodes (6478 knowns and 34,858 unknowns) and 52,137 edges. Compared to original network, it increased 34,939 nodes (i.e., metabolites) with in silico reaction. Through searching against HMDB version 4.0 (released on 2018-12-18), these nodes can be classified as 81 known-known (the compound is included in KEGG database), 405

known-unknowns (the compound is not included in KEGG but included in HMDB), and 34,453 unknown-unknown (the compound is not included in KEGG and HMDB. Consist with MetDNA (v1.3.2), reaction-paired knowns and unknowns have higher MS/MS similarity than non-reaction pairs (Supplementary Fig. 16). The networks can be visualized in Cytoscape (v3.8), and a detailed tutorial of visualization has been provided in Supplementary Note 1.

## Knowledge-guided MS/MS similarity network and annotation propagation

The procedures to curate knowledge-guided MS/MS similarity networks followed our previous MetDNA publication with some modifications. Briefly, seed metabolites were first annotated using the standard MS/MS and RT libraries. The match tolerances were set as MS1 match, 15 ppm; RT match, 20 s; MS2 match, 0.8 (dot product score). The adducts of protonation and deprotonation were used in seed annotation in positive and negative modes, respectively. The seed metabolites were then mapped to KMRN to guide the construction of an MS/MS similarity network with four constraints, including MS1 $m/z$, predicted RT, MS/MS similarity, and metabolic biotransformation. Specifically, seed metabolite-paired knowns/unknowns (constraint 1) were retrieved from KMRN, and their calculated MS1 m/z (constraint 2) and predicted RTs (constraint 3) were matched with experimental values in LC–MS/MS data. Match tolerances of MS1 m/z and RT matches were set as 15 ppm and 30%, respectively. MS2 spectra of qualified peaks were further matched against the surrogated MS2 spectra from seed metabolites (constraint 4). The qualified peaks with dot product score larger than 0.5 or matched fragments more than 4 were linked to the seed metabolites, and their putative structures were assigned from the reaction-paired neighbor metabolites. The RT prediction and MS/MS scoring are consistent with MetDNA. The random forest model was used for RT prediction, where it used seed metabolite RTs and their molecular descriptors for model training. A total of eight and five molecular descriptors optimized in MetDNA were directly used for HILIC and RP systems, respectively. The parameters of the RF model were optimized with tenfold cross-validation via R package "caret" (version 6.0.90). The dot product score with applied square root is used for MS/MS scoring, and no other filtering is applied. Such annotation was propagated in a recursive manner, where newly annotated metabolites were also used as seeds to annotate their neighbor metabolites in LC–MS/MS data. The annotation was terminated until no new metabolites were annotated. A total of 11 and 8 common adducts are considered in annotation propagation in positive and negative modes, respectively (Supplementary Data 7).

## Global peak correlation network

Global peak correlation network is used to recognize all possible ion form peaks in LC–MS data, and further improve the structural assignment (Supplementary Fig. 3). All putatively annotated peaks from knowledge-guided MS/MS network were selected as base peaks, and their co-eluted peaks were extracted from the feature table within ±3 s RT window (composed as a peak group). The recognition of different ion form peaks was performed within each peak group to build a subnetwork, including isotopes, adducts, neutral losses, and in-source fragments (ISF). The detailed procedures are described as follows.

Isotope peaks. The recognition of isotope peak includes the evaluations of mass deviation and intensity ratio. The pairwise m/z distance matrix was first calculated for deviation check. The theoretical m/z of isotopes were calculated as:

$$mz_{isotope} = mz_{base\_feature} + 1.003355 \times N \qquad (1)$$

where $mz_{isotope}$ and $mz_{base\_feature}$ are m/z values for isotopes and base peaks. The N represents the considered number of isotopes with a set of values from 1 to 3 (i.e., [M] to [M+3]). The tolerance for mass

deviation was set as 25 ppm. The deviation of isotope ratio was calculated as:

$$\triangle_{ratio} = \frac{|Int_E - Int_T|}{Int_T} \times 100 \qquad (2)$$

Where $Int_E$ and $Int_T$ are the experimental and theoretical relative intensities[14], respectively. The maximum deviation of the isotope ratio ($\Delta_{ratio}$) was 500% by default.

Adduct and neutral loss peaks. A total of 28 types of adducts and 57 types of neutral losses were considered (Supplementary Data 7). The adduct and neutral loss features are recognized based on mass deviation and feature abundance correlations among samples. The theoretical m/z values of adducts and neutral losses were calculated and matched within each peak group. The tolerance of mass deviation was set as 25 ppm. The feature abundance correlation among samples are calculated between the recognized feature and the base feature, where feature pairs with Pearson correlation coefficient larger than 0.3 are reserved by default. The isotopes of the adduct and neutral loss peaks were also identified using the same approach in "isotope peaks".

In-source fragment peaks. The in-source fragment was retrieved from the MS/MS spectrum of base features and co-eluted MS1 features. The top five intense fragments in the MS/MS spectrum of the base peak were considered as possible in-source fragments and matched with the features in one feature group. The m/z tolerance was set as 25 ppm. The isotopes of in-source fragment features were also recognized following the same approach in "isotope peaks".

As a result, a peak correlation subnetwork of one base peak was constructed by connecting the base peak and different ion form peaks. For all base peaks from the knowledge-guided MS2 similarity network, a list of subnetworks (referred to as "subnetwork list") were generated. All subnetworks were merged as a global peak correlation network. The subnetwork optimization and filtering in the global peak correlation network was performed as follows (Supplementary Fig. 4). Specifically, it contains three steps: (1) check of empirical rules. For each subnetwork in the subnetwork list, we checked whether it quantified the empirical rules (see details in Supplementary Data 7). The disqualified subnetworks were removed from the subnetwork list (e.g., type 1−subnetwork of M175T462_2_[2 M+H]+ in Supplementary Fig. 3); (2) removal of conflict peaks. We checked the conflict base peaks across different subnetworks in the subnetwork list, where conflict base peaks represent the same base peak has different adduct or neural loss annotations (e.g., type 2−subnetworks of M195T69_[M+H]+ and M195T68_[M-H2O+H]+ in Supplementary Fig. 3). To solve the conflict, we reserved the base peak and its subnetwork with the larger size in the subnetwork list; (3) consolidation of redundant ion form peaks. We consolidated different base peaks originated from the same metabolite (e.g., type 3−subnetworks of M153T279_[M+H]+, M170T280_[M+NH4]+, and M135T279_[M-H2O+H]+ in Supplementary Fig. 3). Similarly, the subnetwork with the maximum size was kept in the subnetwork list. Finally, the reserved subnetworks in the subnetwork list were exported as network 3, and related metabolite candidates were also exported.

## Annotation confidence and reporting

Annotated structures are assigned with different confidence levels according to the definition by Metabolomics Standards Initiative (MSI)[51]. The confidence levels were defined as follows: level 1: metabolites annotated using in-house metabolite standards with three orthogonal properties (i.e., MS1 + RT + MS/MS); level 2: metabolites annotated using two orthogonal properties from the standard MS/MS libraries without RT available (i.e., MS1 + MS/MS); level 3.1: known KEGG metabolites annotated with MS1, predicted RT, and surrogate MS/MS spectra (i.e., MS1 + Pred. RT + Surro. MS/MS); level 3.2: unknown structures annotated with MS1, predicted RT, and surrogate

MS/MS spectra (i.e., MS1 + Pred. RT + Surro. MS/MS). It should be noted that not all metabolites annotations obtained from MS1 and MS/MS matches are level 2 in biological samples, especially for metabolites such as hexenoic acid isomers. The confidence level should be adjusted properly in specific situations.

For each feature, all candidates were ranked with a total score ($S_{total}$), which was calculated as Eq. (3):

$$S_{total} = S_{iden} + S_{confidence} \qquad (3)$$

where $S_{iden}$ and $S_{confidence}$ represent the identification score and confidence score, respectively.

The identification score was calculated as Eq. (4):

$$S_{iden} = W_{m/z} \times S_{m/z} + W_{RT} \times S_{RT} + W_{MS/MS} \times S_{MS/MS} \qquad (4)$$

where $S_{m/z}$, $S_{RT}$, and $S_{MS/MS}$ are $m/z$ match, RT match, and MS/MS match scores, respectively. These scores are calculated using the method as MetDNA. The $W_{m/z}$, $W_{RT}$, and $W_{MS/MS}$ are weights for the $m/z$ match, RT match, and MS/MS match scores, and set as 0.25, 0.25, and 0.5, respectively.

The confidence score was calculated as follows Eq. (5):

$$S_{confidence} = \begin{cases} 3, & level\ 1 \\ 2, & level\ 2 \\ 1, & level\ 3.1/3.2 \end{cases} \qquad (5)$$

For each feature, annotation candidates with the highest confidence level were reported. If multiple annotations with the same confidence level, the top 10 ranked candidates using the total score were kept.

## In vitro metabolism experiment with human liver S9 fraction

We experimentally incubated a mixture of 46 common metabolites (46std_mix) with the human liver S9 fractions for 24 h. The 46std_mix solution was prepared using the concentrations provided in Supplementary Data 2, and stored at −80 °C before incubation. The incubation followed the previously reported protocol[72] with minor modifications. Before the experiment, 50 µL of pooled human liver S9 fraction solution (H0610.S9) was diluted into 500 µL using water. The NADPH regenerating system was reconstituted with the addition of 3.5 mL of water to make a final volume of 5 mL. The 4× cofactor stock was freshly prepared with the following composition: 10 mM UDPGA, 2 mM GSH, 2 mg/ml PAPS, 0.1 mM acetyl-CoA, and NADPH regenerating system (1 mM NADP, 5 mM glucose-6-phosphate, 1 unit glucose-6-phosphate dehydrogenase). The incubation was performed in the 1.5 mL of Eppendorf centrifuge tube. In each tube, 30 µL of S9 fraction, 30 µL of Tris buffer (0.2 M; pH 7.5; 2 mM MgCl$_2$), and 30 µL of 46std_mix solution were first pooled. To start the reaction, 30 µL of 4× cofactor stock was added, and the incubation was carried out at 30 °C for 24 h. 360 µL of MeOH:ACN (1:1, v-v) were added to terminate the reaction and extract the metabolites. For metabolite extraction, the sample was incubated at −20 °C for 1 h to facilitate protein precipitation. After the incubation, samples were centrifuged at 13,000 rpm and 4 °C for 15 min. The supernatant was taken out and evaporated to dryness at 4 °C. The samples were reconstituted with 120 µL of ACN/H$_2$O (v:v, 1:1) and vortexed for 30 s and sonicated for 10 min at 4 °C water bath. Finally, the samples were centrifuged for 15 min at 17,000 × $g$ and 4 °C. The supernatant was taken into the sample vial for the LC−MS experiment. Finally, the annotated known and unknown metabolites from incubated 46mix_std samples were validated using multiple strategies. For known metabolite annotations, chemical standards (MS1 + RT + MS/MS, level 1) and public spectral database (MS1+ MS/MS, level 2) were used. Public spectral libraries included NIST17, SonnenburgLabLib, Metlin, MassBank, Fiehn HILIC library, and GNPS library. For validation of unknown metabolites, 3 different in silico MS/MS tools were used, including MetFrag (version 2.4.5-CL), CFM-ID (version 2.4), and MS-FINDER (version 3.24).

## Preparation of biological samples

The biological samples were extracted following our published protocols[73]. In brief, NIST urine samples were thawed at 4 °C on ice. Then 150 µL of urine samples were taken and transferred into a centrifuge tube, and 600 µL of MeOH were added to extract the sample. After vortexed for 30 s and sonicated for 10 min at 4 °C in a water bath, the samples were incubated for 1 h at −20 °C to facilitate protein precipitation. After the incubation, the samples were further centrifuged for 15 minutes at 17,000 × $g$ and 4 °C. The supernatant was collected and evaporated to dryness at 4 °C. The dry extracts were then reconstituted into 150 µL of ACN:H$_2$O (1:1, v/v), followed by sonication at 4 °C for 10 min, and centrifuged at 17,000 × $g$ and 4 °C for 5 min to remove the insoluble debris before LC−MS/MS analysis. For NIST plasma, 100 µL of NIST plasma was extracted using 400 µL of a solvent mixture of MeOH:ACN (1:1, v/v) in the centrifuge tube, and then the mixture was vortexed for 30 s and sonicated for 10 min at 4 °C water bath. The rest of the procedure was the same as described for the NIST urine sample. BV2 cell lines were originally purchased from ATCC with product number CRL-3265. BV2 cells, it was plated in 6-cm dishes at 2000,000 cells/dish, and cultured in DMEM medium containing FBS (10%) and penicillin/streptomycin (1%). The culture medium was quickly removed, and the cells were washed with cold PBS twice. The cell dishes were placed on dry ice and the metabolite extraction solution (ACN/MeOH/H$_2$O = 2/2/1, v/v/v, 1 mL) was added to the dishes to quench the metabolism. The extraction solution was pre-cooled at −80 °C for 1 h prior to the extraction. The plates were then incubated at −80 °C for at least 40 min. The cell contents were scraped and transferred to a 1.5 mL Eppendorf tube. The samples were vortexed for 1 min and centrifuged for 10 min at 17,000 × $g$ and 4 °C to precipitate the insoluble materials. The rest of the procedure was the same as described for the NIST urine sample.

## LC−MS/MS

LC−MS analysis was performed using a UHPLC system (1290 series; Agilent Technologies, USA) coupled with a quadruple time-of-flight mass spectrometer (TripleTOF 6600, SCIEX). A Waters ACQUITY UPLC BEH Amide column (particle size, 1.7 µm; 100 mm (length) × 2.1 mm (i.d.)) was used for the LC separation and the column temperature was kept at 25 °C. Mobile phase A was 25 mM ammonium hydroxide (NH$_4$OH) and 25 mM ammonium acetate (NH$_4$OAc) in water, and B was ACN for both the positive and negative modes. The flow rate was 0.3 mL/min and the gradient were set as follows: 0−1 min: 95% B, 1−14 min: 95% B to 65% B, 14−16 min: 65% B to 40% B, 16−18 min: 40% B, 18−18.1 min: 40% B to 95% B, and 18.1−23 min: 95% B. The injection volume was 2 µL. The data acquisition was operated using the information-dependent acquisition (IDA) mode. The source parameters were set as follows: ion source gas 1 (GAS1), 60 psi; ion source gas 2 (GAS2), 60 psi; curtain gas (CUR), 30 psi; temperature (TEM), 600 °C; declustering potential (DP), 60 or −60 V in positive or negative modes, respectively; and ion spray voltage floating (ISVF), 5500 or −4000 V in positive or negative modes, respectively. The TOF MS scan parameters were set as follows: mass range, 60−1200 Da; accumulation time, 200 ms; and dynamic background subtract, on. The product ion scan parameters were set as follows: mass range, 25−1200 Da; accumulation time, 50 ms; collision energy, 30 or −30 V in positive or negative modes, respectively; collision energy spread, 0; resolution, UNIT; charge state, 1 to 1; intensity, 100 cps; exclude isotopes within 4 Da; mass tolerance, 10 ppm; the maximum number of candidate ions to monitor per cycle, 6; and exclude former target ions, for 4 s after two occurrences. Analyst TF 1.6 software was used for LC−MS/MS data acquisition.

## Evaluation of peak annotation accuracy

The peak annotation contains metabolite annotation and their associated ion form annotation (Supplementary Fig. 5). For each data set, the MS1 peak table was first processed and generated using XCMS. Then, a total of 242 metabolites were identified through matching with MS1 (≤25 ppm), experimental MS/MS spectra (DP score ≥0.8), and standard RT values (RT error ≤30 s) in an in-house chemical standard library. These identifications are MSI level 1 identification. Then, these metabolites were manually confirmed by two independent mass spectrometrists by extracting their EIC and MS/MS spectra from the raw data. As a result, a list of manually checked metabolites were obtained. These metabolites were used as base peaks. Their related ion form peaks were also manually assigned in the peak table, and further verified by extracting their co-eluted EICs, profiles of ion intensities, and MS/MS intensities in raw data. Finally, a total of 3451 metabolic peaks (2130 from positive mode and 1321 from negative mode) originated from 242 metabolites were labeled as metabolites, isotopes, adducts, and in-source fragments (Supplementary Data 1). Among them, some metabolites were measured in both positive and negative modes. All of the 3451 metabolic peaks were used to evaluate the accuracies of peak annotation using MetDNA1 and KGMN (MetDNA2). The first layer of InChIKey of metabolite was used to evaluate its accuracy. The correct annotation is defined as the software returned correct metabolites within the top three candidates.

## Repository-mining and validation of recurrent unknowns

Recurrent unknowns were obtained by searching MS/MS spectra of putative unknown metabolites against GNPS/MassIV repository-mining via MASST[50] (http://gnps.ucsd.edu). The data were filtered using the default parameters in GNPS. The mass tolerance for precursor ion and fragment ion was set as 0.01 Da. The library spectra were filtered in the same manner as the input data. All matches between input spectra and library spectra were required to have a score above 0.7 and at least two matched peaks. The labels of organs and species are manually added according to the description of projects and their meta information (Supplementary Data 6). The MASST search is based on MS1 and MS/MS similarity without considering the retention time. We named a spectrum using the feature name because this is consistent with the feature list in the KGMN analysis. A detailed tutorial on combining KGMN and repository-mining has been provided in Supplementary Note 2.

Recurrent unknown metabolites are validated with chemical standards through chemical or enzymatic syntheses. O-sulfotyrosine: The O-sulfotyrosine was synthesized by MuJin Biotech Inc, Shanghai, China. The experimental protocols for synthesis are below: 500 mg of Fmoc-Tyr(SO$_3$Na)-OH was added to 25% piperidine/DMF solution at room temperature and stirred for 0.5 h under nitrogen protection. Saturated NaCl solution was added to the mixture, followed by ethyl acetate, and extracted three times in a separating funnel, and then the organic fraction was rotary evaporated to obtain 220 mg of a viscous liquid. After dissolved in water, it was purified by HPLC preparative chromatography (0–20%, acetonitrile/water; flow rate at 25 ml/min; room temperature) to obtain O-sulfotyrosine sodium salt, which was lyophilized to obtain 100 mg of powder. The percent yield was 35.7%. The O-sulfotyrosine structure was confirmed by nuclear magnetic resonance spectroscopy ($^1$H NMR, 400 MHz, Methanol-d4), δ(ppm) = 7.26 (s, 4H), 3.78 – 3.49 (m, 1H), 3.03 (t, J = 5.5 Hz, 2H). 4-hydroxyhippuic acid was synthesized and purchased from Sunway (Shanghai, China) with product number CB03526. The chemical structure was confirmed by nuclear magnetic resonance spectroscopy ($^1$H NMR, 400 MHz, DMSO-d6), δ(ppm) = 12.53 (s, 1H), 10.03 (s, 1H), 8.56 (t, J = 5.9 Hz, 1H), 7.73 (d, J = 8.0 Hz, 2H), 6.81 (d, J = 8.0 Hz, 2H), 3.87 (d, J = 5.8 Hz, 2H). 3-hydroxyhippuic acid was synthesized and purchased from Macklin (Shanghai, China) with product number H881672. The chemical structure was confirmed by nuclear magnetic resonance spectroscopy ($^1$H NMR, 400 MHz, DMSO-d6), δ(ppm) = 12.56 (s, 1H), 9.69 (s, 1H), 8.71 (t,

J = 5.9 Hz, 1H), 7.41 – 7.09 (m, 3H), 6.92 (dt, J = 6.9, 2.3 Hz, 1H), 3.88 (d, J = 5.8 Hz, 2H). The NMR spectra were analyzed using MestReNova (v9.0.1) software. NMR and HRMS data for synthesized compounds have been deposited at https://doi.org/10.5281/zenodo.7233722, Protocatechuic acid-3-O-sulfate and 3-hydroxybenzoic acid-3-O-sulfate were in-house synthesized using enzymatic reaction (S9 fraction incubation) of their precursors protocatechuic acid and 3-hydroxybenzoate, respectively.

## Validations of unknowns using in silico MS/MS tools

For validation of unknown metabolites in biological samples, three different in silico MS/MS tools were used, including MetFrag (version 2.4.5-CL), CFM-ID (version 2.4), and MS-FINDER (version 3.24). The format of imported data and parameters were adjusted according to requirements of each tool. The detail parameters were kept the same as in our previous publication[62]. A detailed tutorial on combining KGMN and in silico MS/MS has been provided in Supplementary Note 3.

## Reporting summary

Further information on research design is available in the Nature Portfolio Reporting Summary linked to this article.

## Data availability

All the metabolomics data sets described in our study can be downloaded from the MetDNA2 website (http://metdna.zhulab.cn/). The raw data files of NIST human urine, NIST human plasma, and BV2 cells can be accessed at the National Omics Data Encyclopedia under Accession Code OEP003157. The raw data of in vitro metabolism can be accessed at National Omics Data Encyclopedia under Accession Code OEP003284. The raw data of fruit fly heads are available at MetaboLights under Accession Code MTBLS612 and MTBLS615. The raw data of mouse liver are available at MetaboLights under Accession Code MTBLS601 and MTBLS606. Raw NMR data and HRMS data for synthesized compounds have been deposited to Zenodo [https://doi.org/10.5281/zenodo.7233722]. Supplementary Data 1–7 can also be accessed at Zenodo (https://doi.org/10.5281/zenodo.7089991). The knowledge-based metabolic reaction network and the network files of the supplementary figures are provided as Supplementary Data 8 and 9, respectively. Source data are provided with this paper.

## Code availability

KGMN was mainly developed using R and is executed in MetDNA2. The source code of MetDNA2 is provided in GitHub [https://github.com/ZhuMetLab/MetDNA2] and Zenodo [https://doi.org/10.5281/zenodo.7230249][74]. The completed functions are provided in the MetDNA2 webserver [http://metdna.zhulab.cn/] via free registration. The detailed tutorial is provided in the MetDNA2 webserver and GitHub. The source code for the in silico MS/MS validation (MetDNA2InSilicoTool) are provided in GitHub [https://github.com/ZhuMetLab/MetDNA2InSilicoTool] and Zenodo [https://doi.org/10.5281/zenodo.7233184][75]. The source code of the multilayer network visualization (MetDNA2Vis) were provided in GitHub [https://github.com/ZhuMetLab/MetDNA2Vis] and Zenodo [https://doi.org/10.5281/zenodo.7233189][76].

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

## Acknowledgements

The work is financially supported by the National Natural Science Foundation of China (22022411 to Z.-J.Z.), the Strategic Priority Research Program of the Chinese Academy of Sciences (XDB39050700 to Z.-J.Z.), Major Research Plan of National Natural Science Foundation of China (92057114 to Z.-J.Z.), Science and Technology Commission of Shanghai Municipality (21JC1405902 to Z.-J.Z.), Shanghai Municipal Science and Technology Major Project (2019SHZDZX02 to Z.-J.Z.), and Shanghai Key Laboratory of Aging Studies (19DZ2260400 to Z.-J.Z.).

## Author contributions

Z.-J.Z. and Z.Z. conceived the idea and designed the algorithm and software. Z.Z. developed the KGMN workflow and MetDNA2 package. M.L. performed the sample preparation, data acquisition, and data processing. M.L. and Z.Z. contributed to the tutorial of MetDNA2 webserver. H.Z. contributed to the MASST search and labeling. M.L. and Z.Z. performed the data analysis. Y.Y. contributed to the deployment of MetDNA2 webserver. Y.C. contributed to the preparation of manuscript. Z.Z. and M.L. tested and debugged the program and webserver. Z.-J.Z. and Z.Z. wrote the manuscript. Z.-J.Z. supervised the project.

## Competing interests

The authors declare no competing interests.
