## [Peer Review File · Nature Communications]

REVIEWER COMMENTS

Reviewer #1 (Remarks to the Author):

The authors did a nice job responding to the first round of comments. I do not need to see the final revised version. Some minor clarifications may be warranted as pointed out below.

Regarding the comment. “Most importantly, unlike other peak correlation network approaches such as CAMERA and IIMN, our global peak correlation network can effectively optimize putative metabolite annotations from the first two-layer networks” it seems that here optimization is really filtering for metabolism based interpretation of the data (here they show an example based on the KEGG metabolic pathways and candidate metabolic transformations). In that case “optimizing” is not the right word and it should be filtering. Thus it seems that network optimization is a network filtering for likely related molecules that originate from metabolism, is this a correct interpretation? If not, it would be great to get more clarity what is being optimized in this case? Suggestion remove the term optimization and use filtering instead.

Rev 2 brought up a good point wrt to the term abiotic as it is used here essentially to indicate anything that is not protonated – this is not quite correct. Abiotic is often referred to as non-biological and thus I would change this nomenclature in the manuscript. We also struggled with this nomenclature for IIMN and ended up referring to them as “different ion forms” instead of abiotic.

RE Table S16: The statement that GNPS and NetID are just 1-2 layers is not quite right. For GNPS I have seen as many as six network layers (just not described that way)

IIMN has matches against a reference library (or in silico), it has MS/MS similarity and peak shape similarity, often additional layers such as in silico predictions, delta masses, explanations of delta masses (e.g. <https://www.pnas.org/doi/10.1073/pnas.1710248114>). Mass2Motif etc are added. NetID has three on its own. It is just that they are not described as multi-layers in those papers when they are. Suggestion, I would remove the table altogether as it is not an accurate description.

It is exciting that another ecosystem is being developed that can be leveraged to perform modern metabolomics analysis and is becoming available to the community. So great to see. Best, Pieter and good luck with the final submission.

Reviewer #2 (Remarks to the Author):

The authors have submitted a revision of the article originally submitted to NBT, and addressed the majority of issues me and other reviewers have raised.

- The lack of references in the introduction and comparison to existing approaches has been addressed very well.

- The Supplementary tutorials are really nicely done and greatly help to understand the procedures.

- I would still prefer to not see the term "abiotic feature" being used. In ChEBI

the compound "pyridinestrone-3-carboxylic acid" is described as "... An abiotic metabolite in the 4,5-seco pathway" [CHEBI:138285] so it still has real biochemistry in it.

The term abiotic was not used in 10.1007/s00216-020-03019-3 nor 10.1093/bioinformatics/btz207, 10.1021/acs.analchem.8b03126 nor 10.1146/annurev-biochem-061516-044952, all of which are recent, from different groups and deal with ion annotation in LC/MS.

When specifically searching for adducts, isotopes AND the term abiotic the article

10.1021/acs.analchem.9b02354 came up, where "abiotic" was only mentioned once in the bibliography

containing a paper on "... Metabolomics-Guided Elucidation of Plant Abiotic Stress Responses..."

So the term abiotic remains confusing, please avoid it altogether. Indeed, having an agreed term for these analytical artefacts would be great.

- The supporting data was made more findable by submission to zenodo and GitHub.

- My biggest concern was that I had difficulties understanding some details of the approach and how to reproduce the presented analyses and figures.

- l151: How was the sub-network removed ? + l178: "no manual analyses"

=> The paragraph around original l151 (now l184) is unchanged. Authors have removed the emphasis "no manual analyses were required", leaving the rest of the paragraph unchanged. "... global peak correlation network-based analysis, which optimizes annotations from network 1 and network 2."

Here, it is still unclear to me which step in the procedure actually removes that sub-network. Is it just the boolean "AND" that edges/annotations appear in both network layers ?

- Most of my difficulties to understand the analysis of the KGMN annotation accuracy have been eliminated.

- "Original l165: In the manually curated Supplemental data 1 find 183 unique values in column inchikey1 from 2130 features, not the 242 and 3451 reported in the manuscript."

Thanks for the clarification, that was not clear from the text alone.

I did check all sheets, but didn't realise I have to sum features up, or in the case of the metabolites also have to eliminate duplicates across polarities.

My suggestion:

"Finally, 2,130 (pos) + 1,321 (neg) for a total of 3,451 metabolic peaks originated from 242 metabolites (some measured both pos/neg mode) were"

or maybe phrased a bit more elegantly than my quick attempt.

- I also now understand that Dataset6 better.

The Dataset6 / Sheet 3 / Column A filenames only need a slight modification

and can be converted to

ftp://massive.ucsd.edu/MSV000085879/ccms_peak/peak/012_DaLi_H9M9_H_MCT_R4_neg_17.mz
ML

although it is a bit confusing where the prefix "f.MSV000085879/ccms_peak/..."

in the table comes from. Consider adding the information on the FTP site

and in case the "f.MSV..." was added only locally, consider to remove that

in the Dataset6 tables.

- I still have a problem with "The 8 datasets and 45 mzML files can be found

in sheet 1 in Supplementary Data 6."

I observe M262T526 in Sheet 1 (grep M262T526 'Supplementary Data 6-sheet1.csv' | wc -l)

from 10 different clusters (11496 11528 435 5708 7340 765 87207 87214 87230 87234)

in 7 different studies (MSV000081364 MSV000081463 MSV000081957 MSV000082493
MSV000084062 MSV000084107 MSV000086207)

from 41 different files (checking column 5). BTW, file names alone must not be counted,

as both E12_2.mzML and E12_3.mzML appear in two separate studies.

Still, I couldn't reproduce the numbers 8 and 45.

- "it would not be easy to perform such an overrepresentation analysis for such
a comprehensive repository"

=> I totally agree. That would be a major project itself. If you didn't do

a statistical overrepresentation analysis, please don't call it "enriched".

- Response to Reviewer #3: I really like the performed "a similar evaluation through randomly removing 25% level 1 seed metabolites" done by the authors.

Minor comments:

=====

- l608: How was the RT prediction done ? Which was the training data, what RT accuracy was achieved in a proper cross validation ?

How is re-training performed when the analysis is run on a different analytical setup?

- Supplementary Figure 14: How is the metabolite likeness score calculated ?

Please do cite relevant literature.

Tiny comments:

=====

l109: along metabolic networks (or: along the metabolic network)

l147: supp fig 3 ?

l158: outputted -> qualified / given

l214: Generation of ...

l806: at Zenodo

- Suppl Fig 1: The largest subnetwork is *shown* here

- Suppl Fig 2: For the future: such data might be better visualised via a histogram/density, with the mean and quantiles given in the caption. No update necessary here.

- Suppl Table 1: Links don't have a URL (tested in Firefox, Chrom and Linux/evince).

- Suppl Tutorial 2: No link for "instructions of MASST can be found in GNPS document."

- Actually, none of the links in 376515_1_supp_6700238_rqppkr.pdf seem to work.

Could that be an issue with the Nature Processing Systems ?

Reviewer #3 (Remarks to the Author):

The authors did substantially improve their manuscript and work on many aspects as compared to the previous version - thanks for considering the reviewer remarks and suggestions. The manuscript is now much better referenced by citing relevant literature and it is good to observe that much more of the workflow and data is now accessible and thus reproducible - this is likely to facilitate its uptake by the community. The additional benchmarking efforts included so far are also appreciated, alongside a bit more clarity on the workflow. However, this also raised a number of new questions. Please find a set of items that the authors should consider before publication, some of them being a continuation of a previous point raised by one of the reviewers. This reviewer would like to thank the authors for their contribution to the metabolomics field - their efforts are much appreciated.

Scope (Title/Abstract)  the authors agree with the reviewer that the current implementation of KGMN does not necessarily work for all sample types, and they added one line (P18) to reflect this aspect, but did not do so in the title or abstract. Whilst this reviewer agrees with the authors on the potential of their tool (once more "knowledge" is fed into the "knowledge-guided" part, the scope of the tool will expand), the current implementation has clear limitations, that are not acknowledged in the title or abstract. Now, it seems like the approach can be used on any dataset, whereas in practice, this reviewer is convinced (based on the manuscript) that it will work best for humane samples and sample origins like E. coli and yeast, for which a substantial number of molecules are reported, and (bio)reaction rules are described. It is of note that even in those sample types, not all metabolites will be annotated following a knowledge-driven approach (alone), as xenobiotics can be present in bio fluids that are not part of the "known metabolite repertoire" and cannot be connected to from there. To this reviewer, the authors have to pay attention to this aspect in the title and abstract to aid the reader in finding the best tool for the job they have at hand. For extracts from the environment, and non-model plants, fungi, and bacteria, the current KGMN may not be the best approach - based on the current knowledge that goes into the approach. In the discussion, the authors could extend the section on P18 a bit by including the current knowledge (structures, pathways) that goes into KGMN, and how they foresee that the mentioned tools and database will help to extend the applicability of KGMN. A next version of the tool may encompass a wider applicability, which the authors can then mention in the abstract.

Benchmarking  the authors argue that it is not possible to benchmark their proposed approach with NetID because the aim of the two tools is too different. However, MetDNA (their own previous tool) also had a different aim and it could be argued that benchmarking against MetDNA would be incorrect as well. Instead of following this line of thought, this reviewer would argue that NetID can be run on some of the datasets and the in this revision introduced level 1 identifications can be used for benchmarking (i.e., does NetID find the same annotations). Based on the abstract (and this is cited: The approach aims to generate, for all experimentally observed ion peaks, annotations that match the measured masses, retention times and (when available) tandem mass spectrometry fragmentation patterns. Peaks are connected based on mass differences reflecting adduction, fragmentation, isotopes, or feasible biochemical transformations. Global optimization generates a single network linking most observed ion peaks, enhances peak assignment accuracy, and produces chemically informative peak-peak relationships, including for peaks lacking tandem mass spectrometry spectra.), NetID uses similar concepts and follows comparable steps, thus likely resulting in comparable results. This reviewer acknowledges that KGMN also has unique features, but a comparison is needed for metabolomics researchers studying the available tools to decide what fits their needs. For example, do NetID and KGMN group the same adduct and artefact features together for level 1 annotations? Now, this remains opaque.

Multilayered Network (across the manuscript)  the authors have put effort in explaining how to export and visualize the three networks as described in the manuscript, but they did not really answer to the raised issue. To this reviewer, it seems that the KGMN approach is more an integrative networking approach (i.e., knowledge-guided integrative network - KGIN approach) as it is still unclear if a multilayered network (i.e., nodes with several types of edges) is actually used and can directly be used for the metabolite annotation analysis. If understood correctly, the knowledge-guided part results in a network of structures linked by reactions; and it can only be integrated (linked) to the data-driven network by matching m/z to formulas, right? These two networks do not share nodes, so the approach "integrates" them automatically and one adds annotations to the other - akin to using BioTransformer on a set of molecular structures, and then MetFrag to add annotations to a molecular network. It is not the purpose of this reviewer to start a "semantic dispute", but it would be good to think about the naming of the approach and also explain in the main text what the authors actually mean with "multi-layered". This review may be of interest to the authors within this respect: <https://bdataanalytics.biomedcentral.com/articles/10.1186/s41044-020-00046-0>.

P5 L91 - add a reference to the here introduced repository-scale mining. Furthermore, briefly explain in a sentence how a connection was made to aid the reader in understanding the proposed workflow.

P6 L133-134 "4 constraints"  the authors provided more information about each of the three networks they use. They highlight the "4 constraints" as being unique for their KGMN approach. However, it is not intuitive how each of these constraints aid in finding metabolite annotations.

Therefore, more clarity is needed for each constraint. For example, how does "MS1 m/z" adds? Is there a gap on the maximum m/z difference? Etc.

P7 L155-157 "global peak correlation network"  following previous remarks, now the authors state that their implementation of feature grouping (i.e., global peak correlation network (network 3), akin to CAMERA and IIN) is unique. However, they do not explain or highlight in the main text how it is unique. To this reviewer, not everything needs to be novel/unique to make an approach useful; however, when this is claimed, it has to be explained. Please elaborate and clearly explain what is unique. The authors also state here that it "effectively optimizes putative metabolite annotations" - did the authors compare their approach to, e.g., CAMERA? Is it much better? Again, this is not a necessity for this reviewer, but it is good practice to build on (and implement) good functioning packages and only build new ones when it is needed. Also, when it becomes clearer which aspects improved the results, complementary tools could also adopt them.

P17 L401-402 "Hooft group"  based on the cited reference, it is likely that the authors mean the "Van der Hooft Computational Metabolomics Group". If so, please adjust.

GitHub repository - please note that on the main page, it is noted that the sample csv file needs "three" columns, whereas it should be "two" (according to the tutorials, and to the two column names that are specified).

Response to the reviewers:

The authors would like to thank the reviewers for the helpful comments. We feel these comments have strengthened the manuscript considerably.

Reviewer #1:

Remarks to the Author: *“The authors did a nice job responding to the first round of comments. I do not need to see the final revised version. Some minor clarifications may be warranted as pointed out below.”*

Ans: We appreciate the reviewer’s positive comments towards publication. We feel these comments have significantly strengthened the manuscript.

Comment #1: *“Regarding the comment. “Most importantly, unlike other peak correlation network approaches such as CAMERA and IIMN, our global peak correlation network can effectively optimize putative metabolite annotations from the first two-layer networks” it seems that here optimization is really filtering for metabolism bases interpretation of the data (here they show an example based on the KEGG metabolic pathways and candidate metabolic transformations). In that case “optimizing” is not the right word and it should filtering. Thus it seems that network optimization is a network filtering for likely related molecules that originate from metabolism, is this a correct interpretation? If not, it would be great to get more clarity what is being optimized in this case? Suggestion remove the term optimization and use filtering instead.”*

Ans: We agree with the reviewer’s suggestion. We have revised the word “optimize” as “filter” in the revised manuscript.

Comment #2: *“Rev 2 brought up a good point wrt to the term abiotic as it is used here essentially to indicate anything that is not protonated – this is not quite correct. Abiotic is often referred to as non-biological and thus I would change this nomenclature in the manuscript. We also struggled with this nomenclature for IIMN and ended up referring to them as “different ion forms” instead of abiotic.”*

Ans: Thanks a lot for the reviewer’s suggestion. The term “abiotic peaks” have been modified as “different ion forms” in the revised manuscript.

Comment #3: *“RE Table SI6: The statement that GNPS and NetID are just 1-2 layers is not quite right. For GNPS I have seen as many as six network layers (just not described that way). IIMN has matches against a reference library (or in silico), it has MS/MS similarity and peak shape similarity, often additional layers such as in silico predictions, delta masses, explanations of delta masses (e.g.*

https://www.pnas.org/doi/10.1073/pnas.1710248114). Mass2Motif etc are added. NetID has three on it own. It is just that they are not described as multi-layers in those papers when they are. Suggestion, I would remove the table altogether as it is not an accurate description. ”

Ans: We agree with the reviewer's comment. We have removed the Table S6 in the revised manuscript.

Comment #4: *“It is exciting that another ecosystem is being developed that can be leveraged to perform modern metabolomics analysis and is becoming available to the community. So great to see. Best, Pieter and good luck with the final submission.”*

Ans: Thanks a lot for the reviewer's positive comment.

Reviewer #2:

Remarks to the Author: *"The authors have submitted a revision of the article originally submitted to NBT, and addressed the majority of issues me and other reviewers have raised."*

Ans: Thanks a lot for the reviewer's positive comments towards publication. We feel your constructive comments strengthen the manuscript considerably.

Comment #1: *"The lack of references in the introduction and comparison to existing approaches has been addressed very well. "*

Ans: Thanks a lot for the reviewer's comment.

Comment #2: *"The Supplementary tutorials are really nicely done and greatly help to understand the procedures.."*

Ans: Thanks a lot for the reviewer's positive comment.

Comment #3: *"I would still prefer to not see the term "abiotic feature" being used. In ChEBI the compound "pyridinestrone-3-carboxylic acid" is described as "... An abiotic metabolite in the 4,5-seco pathway" [CHEBI:138285] so it still has real biochemistry in it. The term abiotic was not used in 10.1007/s00216-020-03019-3 nor 10.1093/bioinformatics/btz207, 10.1021/acs.analchem.8b03126 nor 10.1146/annurev-biochem-061516-044952, all of which are recent, from different groups and deal with ion annotation in LC/MS. When specifically searching for adducts, isotopes AND the term abiotic the article 10.1021/acs.analchem.9b02354 came up, where "abiotic" was only mentioned once in the bibliography containing a paper on "... Metabolomics-Guided Elucidation of Plant Abiotic Stress Responses..." So the term abiotic remains confusing, please avoid it altogether. Indeed, having an agreed term for these analytical artefacts would be great."*

Ans: Thanks a lot for the reviewer's suggestion. As suggested by reviewer #1, we have revised the "abiotic peaks" as "different ion forms" in the revised manuscript.

Comment #4: *"The supporting data was made more findable by submission to zenodo and GitHub."*

Ans: Thanks a lot for the reviewer's comment.

Comment #5: *"My biggest concern was that I had difficulties understanding some details of the approach and how to reproduce the presented analyses and figures."*

I151: How was the sub-network removed ? + I178: "no manual analyses" => The paragraph around original I151 (now I184) is unchanged. Authors have removed the emphasis "no manual analyses were required", leaving the rest of the paragraph unchanged. "... global peak correlation network-based analysis, which optimizes annotations from network 1 and network 2." Here, it is still unclear to me which step in the procedure actually removes that sub-network. Is it just the boolean "AND" that edges/annotations appear in both network layers?"

Ans: Thanks a lot for the reviewer's comment. In the revised manuscript, we added a flowchart and rewrite the method part to help the reviewer and readers to understand the workflow related to global peak correlation network (**Supplementary Fig. 4 in the revised manuscript**). Briefly, annotated peaks in the network 2 were firstly selected as base peaks. Then, each base peak and its co-eluted peaks ($\pm 3s$ RT window) were combined to generate the peak group. Next, different ion forms (abiotic peaks in the previous manuscript) were annotated within each peak group to construct a subnetwork. As a result, a list of subnetworks (referred as "subnetwork list" in below) were generated. All subnetworks were merged as a global peak correlation network. Then, the subnetwork optimization and filtering in the global peak correlation network was performed as follows. First, for each subnetwork in subnetwork list, we checked whether it quantified the empirical rules (see details in **Supplementary Data 7**). The disqualified subnetworks were removed from subnetwork list (e.g., type 1 – subnetwork of M175T462_2_[2M+H]⁺ in **Supplementary Fig. 3**). Second, we checked the conflict base peaks across different subnetworks in the subnetwork list, where conflict base peaks represent the same base peak have different adduct or neutral loss annotations (e.g., type 2 – subnetworks of M195T69_[M+H]⁺ and M195T68_[M-H₂O+H]⁺ in **Supplementary Fig. 3**). To solve the conflict, we reserved the base peak and its subnetwork with larger size in the subnetwork list. Third, we consolidated different base peaks originated from the same metabolite (e.g. type 3 – subnetworks of M153T279_[M+H]⁺, M170T280_[M+NH₄]⁺, and M135T279_[M-H₂O+H]⁺ in **Supplementary Fig. 3**). Similarly, the subnetwork with the maximum size was kept in the subnetwork list. Finally, the reserved subnetworks in the subnetwork list were exported as network 3, and related metabolite candidates were also exported.

We have added a statement in the main text as follows: "The KGMMN approach enables to optimize and filter metabolite annotation and improve the accuracy of peak assignment through global peak correlation network (see **Methods** and **Supplementary Fig. 3-4**)."

We have also revised the methods about subnetwork optimization and filtering as follow: "For all base peaks from knowledge-guided MS2 similarity network, a list of subnetworks (referred as 'subnetwork list') were generated. All subnetworks were merged as a global peak correlation network. The subnetwork optimization and filtering in the global peak correlation network was performed as follows (**Supplementary Fig. 4**). Specifically, it contains 3 steps: (1) check of empirical rules. For each subnetwork in subnetwork list, we checked whether it quantified the empirical rules (see details in **Supplementary Data 7**). The disqualified subnetworks were removed from subnetwork list (e.g., type 1

– subnetwork of $M175T462_2_2[M+H]^+$ in **Supplementary Fig. 3**); (2) removal of conflict peaks. We checked the conflict base peaks across different subnetworks in the subnetwork list, where conflict base peaks represent the same base peak have different adduct or neutral loss annotations (e.g., type 2 – subnetworks of $M195T69_[M+H]^+$ and $M195T68_[M-H_2O+H]^+$ in **Supplementary Fig. 3**). To solve the conflict, we reserved the base peak and its subnetwork with larger size in the subnetwork list; (3) consolidation of redundant ion form peaks. We consolidated different base peaks originated from the same metabolite (e.g. type 3 – subnetworks of $M153T279_[M+H]^+$, $M170T280_[M+NH_4]^+$, and $M135T279_[M-H_2O+H]^+$ in **Supplementary Fig. 3**). Similarly, the subnetwork with the maximum size was kept in the subnetwork list. Finally, the reserved subnetworks in the subnetwork list were exported as network 3, and related metabolite candidates were also exported. “

Supplementary Figure S4. Flowchart for the optimization and filtering of subnetworks in the global peak correlation network.

Comment #6: “Most of my difficulties to understand the analysis of the KGMN annotation accuracy have been eliminated.”

Ans: Thanks a lot for the reviewer’s comment.

Comment #7: “ ‘Original I165: In the manually curated Supplemental data 1 find 183 unique values in column inchikey1 from 2130 features, not the 242 and 3451 reported in the manuscript. ’ Thanks for

the clarification, that was not clear from the text alone. I did check all sheets, but didn't realise I have to sum features up, or in the case of the metabolites also have to eliminate duplicates across polarities.

My suggestion:

"Finally, 2,130 (pos) + 1,321 (neg) for a total of 3,451 metabolic peaks originated from 242 metabolites (some measured both pos/neg mode) were" or maybe phrased a bit more elegantly than my quick attempt."

Ans: Thanks a lot for the reviewer's suggestion. We have revised this part in the revised manuscript as follows: "Finally, a total of 3,451 metabolic peaks (2,130 from positive mode and 1,321 from negative mode) originated from 242 metabolites were.... Among them, some metabolites were measured in both positive and negative modes."

Comment #8: "I also now understand that Dataset6 better. The Dataset6 / Sheet 3 / Column A filenames only need a slight modification and can be converted to ftp://massive.ucsd.edu/MSV000085879/ccms_peak/peak/012_DaLi_H9M9_H_MCT_R4_neg_17.mzML although it is a bit confusing where the prefix "f.MSV000085879/ccms_peak/..." in the table comes from. Consider adding the information on the FTP site and in case the "f.MSV..." was added only locally, consider to remove that in the Dataset6 tables. "

Ans: We agree with the reviewer's comment. We have removed this confusing prefix in **Supplementary Data 6** in the revised manuscript.

Comment #9: "I still have a problem with 'The 8 datasets and 45 mzML files can be found in sheet 1 in Supplementary Data 6.'

I observe M262T526 in Sheet 1 (grep M262T526 'Supplementary Data 6-sheet1.csv' | wc -l) from 10 different clusters (11496 11528 435 5708 7340 765 87207 87214 87230 87234) in 7 different studies (MSV000081364 MSV000081463 MSV000081957 MSV000082493 MSV000084062 MSV000084107 MSV000086207) from 41 different files (checking column 5). BTW, file names alone must not be counted, as both E12_2.mzML and E12_3.mzML appear in two separate studies. Still, I couldn't reproduce the numbers 8 and 45."

Ans: Thanks a lot for the reviewer careful correction. This is a typo. The numbers in the **Figure 5d** are correct. We have corrected this typo as follows: "We further demonstrated an example for a recurrent unknown peak of M262T526, which was observed in 7 data sets and 41 data files (**Figure 5d**)."

In addition, we have revised the numbers of data files according to reviewer's suggestion as below: "A total of 187 unknowns were recurrent in 351 data sets and 13,418 data files (**Figure 5b** and **Supplementary Data 6**)".

Comment #10: *“it would not be easy to perform such an overrepresentation analysis for such a comprehensive repository’*

=> I totally agree. That would be a major project itself. If you didn't do a statistical overrepresentation analysis, please don't call it 'enriched'”

Ans: We agree with the reviewer's comment. We have revised the term “enriched” as “mainly appeared” in the revised manuscript. The related description has been modified as follows: *“We noticed that recurrent unknown metabolites were mainly appeared in human species such as plasma, serum, and urine, which were the same as sample types tested in this study.”*

Comment #11: *“Response to Reviewer #3: I really like the performed "a similar evaluation through randomly removing 25% level 1 seed metabolites" done by the authors.”*

Ans: Thanks a lot for the comment.

Comment #12: *“Minor comments: =====*

- I608: How was the RT prediction done? Which was the training data, what RT accuracy was achieved in a proper cross validation? How is re-training performed when the analysis is run on a different analytical setup?”

Ans: Thanks a lot for the comment. The procedures of RT prediction were consistent with MetDNA with minor modifications. Briefly, for each dataset, the annotated seed metabolites through matching with the spectral and RT libraries were used as the training data set. Then, the optimized combinations of molecular descriptors in MetDNA were directly used, including 8 and 5 molecular descriptors for HILIC and RP systems, respectively. These molecular descriptors were calculated via R package “rcdk” for each annotated seed metabolite. Finally, a random forest (RF) regression model was constructed to learn the relationship between the experimental RT and the molecular descriptors of each seed metabolite. The parameters of the RF model were optimized with the 10-fold cross validation via R package “caret”. Finally, this RF model was used to predict theoretical RTs for known and unknown metabolites in the metabolic reaction network. We tested this approach using our in-house experimental RT library with a median relative error of 12.2% (independent test set for HILIC separation; n = 156 metabolites).

In revised manuscript, we have added related description as below: *“The random forest model was used for RT prediction, where it used seed metabolite RTs and their molecular descriptors for model training. A total of 8 and 5 molecular descriptors optimized in MetDNA were directly used for HILIC and RP systems, respectively. The parameters of RF model were optimized with 10-fold cross validation via R package “caret” (version 6.0.90).”*

Comment #13: “- *Supplementary Figure 14: How is the metabolite likeness score calculated? Please do cite relevant literature.*”

Ans: Thanks a lot for the reviewer’s comment. The metabolite likeness score is calculated via NP likeness calculator (v2.1, *J. Chem. Inf. Model.* 2008, 48, 1, 68–74 ; <https://doi.org/10.1021/ci700286x>). The relevant literature has been added in revised manuscript.

Comment #14: “Tiny comments: =====

I109: along metabolic networks (or: along the metabolic network)

I147: supp fig 3 ?

I158: outputted -> qualified / given

I214: Generation of ...

I806: at Zenodo”

Ans: Thanks a lot for the reviewer’s correction. These typos have been corrected in the revised manuscript.

Comment #15: ”- *Suppl Fig 1: The largest subnetwork is *shown* here*

- *Suppl Fig 2: For the future: such data might be better visualised via a histogram/density, with the mean and quantiles given in the caption. No update necessary here.*

- *Suppl Table 1: Links don't have a URL (tested in Firefox, Chrom and Linux/evince).*

- *Suppl Tutorial 2: No link for "instructions of MASST can be found in GNPS document."*

- *Actually, none of the links in 376515_1_supp_6700238_rqpk.pdf seem to work.*

Could that be an issue with the Nature Processing Systems?”

Ans: Thanks a lot for the reviewer’s correction. These links may be lost during PDF conversion. We added text links in the revised Supplementary Information and tutorials.

Reviewer #3:

Remarks to the Author: *“The authors did substantially improve their manuscript and work on many aspects as compared to the previous version - thanks for considering the reviewer remarks and suggestions. The manuscript is now much better referenced by citing relevant literature and it is good to observe that much more of the workflow and data is now accessible and thus reproducible - this is likely to facilitate its uptake by the community. The additional benchmarking efforts included so far are also appreciated, alongside a bit more clarity on the workflow. However, this also raised a number of new questions. Please find a set of items that the authors should consider before publication, some of them being a continuation of a previous point raised by one of the reviewers. This reviewer would like to thank the authors for their contribution to the metabolomics field - their efforts are much appreciated.”*

Ans: Thanks a lot for the reviewer's comments and suggestions.

Comment #1: *“Scope (Title/Abstract)  the authors agree with the reviewer that the current implementation of KGMN does not necessarily work for all sample types, and they added one line (P18) to reflect this aspect, but did not do so in the title or abstract. Whilst this reviewer agrees with the authors on the potential of their tool (once more "knowledge" is fed into the "knowledge-guided" part, the scope of the tool will expand), the current implementation has clear limitations, that are not acknowledged in the title or abstract. Now, it seems like the approach can be used on any dataset, whereas in practice, this reviewer is convinced (based on the manuscript) that it will work best for humane samples and sample origins like E. coli and yeast, for which a substantial number of molecules are reported, and (bio)reaction rules are described. It is of note that even in those sample types, not all metabolites will be annotated following a knowledge-driven approach (alone), as xenobiotics can be present in bio fluids that are not part of the "known metabolite repertoire" and cannot be connected to from there. To this reviewer, the authors have to pay attention to this aspect in the title and abstract to aid the reader in finding the best tool for the job they have at hand. For extracts from the environment, and non-model plants, fungi, and bacteria, the current KGMN may not be the best approach - based on the current knowledge that goes into the approach. In the discussion, the authors could extend the section on P18 a bit by including the current knowledge (structures, pathways) that goes into KGMN, and how they foresee that the mentioned tools and database will help to extend the applicability of KGMN. A next version of the tool may encompass a wider applicability, which the authors can then mention in the abstract. “*

Ans: Thanks a lot for the reviewer's comment. We agree with the reviewer's comment that current KGMN has the potential limitation. However, due to the maximum length requirements from the journal (15 words and 150 words for the title and the abstract, respectively), we are unable to include a detailed statement in the title and abstract. If reviewer has any specific suggestions for wording in the title and abstract, we would be happy to take these suggestions. Specifically, to address the reviewer's concern, we have revised some words in the abstract to indicate that our KGMN approach is mainly evaluated on common

biological samples from model organisms as follows: *“Together, the KGMN approach enables efficient unknown annotations, and substantially advances the discovery of recurrent unknown metabolites for common biological samples from model organisms, towards deciphering dark matter in untargeted metabolomics.”*

In addition, we further extended the related discussions in Dis as follows: *“Finally, although current KGMN has been tested in several common biological samples, one needs to be cautious when applying it into some cases like non-model organisms, environmental and exposomics-related profiles (e.g., waste waters, non-model plants, fungi, and bacteria). It is feasible to incorporate databases for these organisms in the future, like Nature Products Atlas⁶⁶ (for bacterial and fungal) and T3DB⁶⁷ (for toxin). Through expanding the knowledge of structures and pathways with wider coverages, it is foreseeable that next versions of KGMN may encompass a wider applicability.”*

Comment #2: *“Benchmarking  the authors argue that it is not possible to benchmark their proposed approach with NetID because the aim of the two tools is too different. However, MetDNA (their own previous tool) also had a different aim and it could be argued that benchmarking against MetDNA would be incorrect as well. Instead of following this line of thought, this reviewer would argue that NetID can be run on some of the datasets and the in this revision introduced level 1 identifications can be used for benchmarking (i.e., does NetID find the same annotations). Based on the abstract (and this is cited: The approach aims to generate, for all experimentally observed ion peaks, annotations that match the measured masses, retention times and (when available) tandem mass spectrometry fragmentation patterns. Peaks are connected based on mass differences reflecting adduction, fragmentation, isotopes, or feasible biochemical transformations. Global optimization generates a single network linking most observed ion peaks, enhances peak assignment accuracy, and produces chemically informative peak-peak relationships, including for peaks lacking tandem mass spectrometry spectra.), NetID uses similar concepts and follows comparable steps, thus likely resulting in comparable results. This reviewer acknowledges that KGMN also has unique features, but a comparison is needed for metabolomics researchers studying the available tools to decide what fits their needs. For example, do NetID and KGMN group the same adduct and artefact features together for level 1 annotations? Now, this remains opaque.”*

Ans: Thanks a lot for the reviewer's comment. We understand the reviewer's concern for benchmark between KGMN and NetID. First of all, we think NetID is a data-driven approach while KGMN is a knowledge-driven approach. Therefore, they have distinct principles for peak annotation. In this revision, to address the reviewer's concern, we have tested NetID with 4 different benchmark datasets in our study (human urine sample; head tissue of fruit fly; positive and negative modes; see **Supporting Table R1**). Specifically, we tested two different formats for data inputs: 1) MS1 peak table only (similar to the setting in Figure 2b in NetID paper); 2) MS1 peak table and experimental MS2 spectra. Other parameters were kept the same as the NetID publication. As we mentioned in the response of last revision, NetID only provides formula annotations of peaks. We compared the annotated formula of MS1 features (including

metabolite peaks and related abiotic peaks) with the benchmark datasets. Unexpectedly, although we have tested different settings and datasets, NetID generally showed very poor performances on our data sets (**Supporting Table R1**). For example, in human urine positive mode data set, 217 out of 425 peaks (51%) had putatively assigned formula, while only 11 out of 271 peaks (5.1%) were correctly assigned by NetID. Even we included MS2 spectra as the input data, the NetID performances were not improved. In addition, we also included our experimental RT library to aid annotation in NetID but failed to improve the accuracy (data not shown). Similar results were also obtained in other datasets (**Supporting Table R1**).

Supporting Table R1. Statistics of NetID based annotations of level 1 peaks in benchmark data.

No.	Benchmark Datasets	Manually curated peaks with level 1 annotations	NetID annotations			
			Input: MS1 peak table		Input: MS1 peak table +MS2 spectra	
			Annotated peaks	Correct annotations	Annotated peaks	Correct annotations
1	Human urine (positive)	425	217	11 (5.1%)	240	15 (6.3%)
2	Human urine (negative)	325	207	17 (8.2%)	216	17 (7.9%)
3	Head tissue of fruit fly (positive)	365	234	4 (1.7%)	256	8 (3.1%)
4	Head tissue of fruit fly (negative)	258	149	0	155	0

We think two reasons may contribute to these results. First, the MS1 peak tables for our benchmark data sets were generated from XCMS, which usually had 20,000~30,000 peaks for common biological samples. As a comparison, NetID paper used the peak table generated from MAVEN with 5000-12000 peaks for common biological samples. In general, XCMS tends to generate noise peaks during the peak detection, and we guess these extra noise peaks may have a significant impact on the integer linear programming algorithm employed in NetID, which affected the annotation accuracy of NetID. Since NetID did not test and have an optimized protocol for XCMS-generate peak table. We could not confirm this point. Second, the original NetID is developed and tested only on data from orbitrap instrument, but not QTOF instruments. Our benchmark data sets were acquired on a QTOF instrument. The instrument different may also have a potential contribution. However, NetID did not provide any instructions regarding to parameter optimization for QTOF data. Therefore, we think NetID might need further optimization, testing and detailed instructions for metabolomics data from different instruments and peak picking software, which could ensure meaningful comparisons with KGMN.

In addition, we also tested the software CAMERA to annotation our benchmark data sets (see the response to **Comment #6** and **Supplementary Fig. 7**). The results indicated that our benchmark data sets are compatible with other software tools. Reasonable annotation performances could be achieved with CAMERA for the benchmark data sets.

Comment #3: *“Multilayered Network (across the manuscript)  the authors have put effort in explaining how to export and visualize the three networks as described in the manuscript, but they did not really answer to the raised issue. To this reviewer, it seems that the KGMN approach is more an integrative networking approach (i.e., knowledge-guided integrative network - KGIN approach) as it is still unclear if a multilayered network (i.e., nodes with several types of edges) is actually used and can directly be used for the metabolite annotation analysis. If understood correctly, the knowledge-guided part results in a network of structures linked by reactions; and it can only be integrated (linked) to the data-driven network by matching m/z to formulas, right? These two networks do not share nodes, so the approach "integrates" them automatically and one adds annotations to the other - akin to using BioTransformer on a set of molecular structures, and then MetFrag to add annotations to a molecular network. It is not the purpose of this reviewer to start a "semantic dispute", but it would be good to think about the naming of the approach and also explain in the main text what the authors actually mean with "multi-layered". This review may be of interest to the authors within this respect: <https://bdataanalytics.biomedcentral.com/articles/10.1186/s41044-020-00046-0>. ”*

Ans: Thanks a lot for the reviewer’s comment and sharing the relevant review article. As mentioned in this review, the term of “multi-layer network” lacks a terminology convention, whereas its interpretation and implementation depend on the subject it serving. In this review, the authors summarized two types of multilayer networks, “node-colored graphs” and “edge-colored graphs”. Briefly, in node-colored graphs, nodes have different aspects or types, and do not share across multilayer networks. In contrast, edge-colored graph shares nodes across different layered networks, while they have different type edges in multilayer networks.

According to the definitions, we think KGMN is an approach which integrates above two types of multi-layer networks. Specifically, the relationship between knowledge-guided reaction network and experimental networks (MS2 similarity network & global peak correlation network) belongs node-colored graphs, where nodes are not directly shared. Node represents a metabolite in knowledge-guided reaction network, while note represents an experimental peak (or feature) in the MS2 similarity network and the global peak correlation network. In addition, the relationship between the MS2 similarity network and the global peak correlation network belongs to the edge-colored graph. They shared nodes but have different edge types. Edge in the MS2 similarity network represents MS2 similarity, while edge in global peak correlation network represents ion form relationship (i.e., the abiotic peaks in the previous version). A recent review (<https://doi.org/10.3389/fmolb.2022.841373>) also termed a similar concept which integrates knowledge and experimental networks as a “multilayered network” approach. We understand that the complexity of our approach that combines and integrates multiple different types of networks may cause the difficulties to understand. We wish we have provided enough explanations to define our approach as a multilayered network approach.

To indicate the relationships between different layer networks, we added related discussions in the Discussion as below: *“The definition of multi-layer network caused one of the reviewers’ discussion during revision. According to a recent review⁵⁸, the term of multi-layer network lacks a terminology convention, where the authors summarized two types of multilayer networks, ‘node-colored graphs’ and ‘edge-colored graphs’ . KGMN is an approach which integrates above two types of multi-layer networks. Specifically, the relationship between knowledge-guided reaction network and experimental networks (MS2 similarity network & global peak correlation network) belongs node-colored graphs, where nodes are not directly shared. Node represents a metabolite in knowledge-guided reaction network, while node represents an experimental peak (or feature) in the MS2 similarity network and the global peak correlation network. In addition, the relationship between the MS2 similarity network and the global peak correlation network belongs to the edge-colored graph. They shared nodes but have different edge types. Edge in the MS2 similarity network represents MS2 similarity, while edge in global peak correlation network represents ion form relationship. A recent review also termed a similar concept which integrates knowledge and experimental networks as a multi-layer network approach²⁵.”*

Comment #4: *“P5 L91 - add a reference to the here introduced repository-scale mining. Furthermore, briefly explain in a sentence how a connection was made to aid the reader in understanding the proposed workflow. “*

Ans: Thanks a lot for the reviewer’s comment. We have added the reference in the revised manuscript, and a sentence to help understanding this workflow as follows: *“Finally, we evaluated these putative unknown metabolites whether were recurrent in similar samples in the metabolomics repository⁵⁰. We successfully discovered 5 unknown metabolites that are absent in common MS/MS libraries through integrating KGMN and the repository-mining.”*

Comment #5: *“P6 L133-134 "4 constraints"  the authors provided more information about each of the three networks they use. They highlight the "4 constraints" as being unique for their KGMN approach. However, it is not intuitive how each of these constraints aid in finding metabolite annotations. Therefore, more clarity is needed for each constraint. For example, how does "MS1 m/z" adds? Is there a gap on the maximum m/z difference? Etc.”*

Ans: Thanks a lot for the reviewer’s comment. We are sorry that we did not describe “4 constraints” clearly in the original manuscript. This process is similar as known metabolite annotation in MetDNA. Specifically, seed metabolites were annotated through matching with MS1 *m/z*, experimental RT and MS2 spectra in the library. For each seed metabolite, its reaction-paired neighbor metabolites (**constraint 1**) were retrieved from the knowledge-guided reaction network. These metabolites were used as possible candidates, and their theoretical MS1 *m/z* (**constraint 2**) and predicted RT (**constraint 3**) were

calculated with their formula and chemical structures. The calculated MS1 m/z and RT values were matched to unannotated peaks in the peak table. Match tolerances of MS1 m/z and RT matches were set as 15 ppm and 30%, respectively. MS2 spectra of qualified peaks were further matched against the surrogated MS2 spectra from seed metabolites (**constraint 4**, dot product ≥ 0.5 or matched fragments ≥ 4). Taken together, a total of 4 constraints ensured to select potential peaks with high accuracy for the following annotation propagation.

We have added a statement in the main text, and revised the methods as follows: *“Specifically, seed metabolite-paired knowns/unknowns (constraint 1) were retrieved from KMRN, and their calculated MS1 m/z (constraint 2) and predicted RTs (constraint 3) were matched with experimental values in LC-MS/MS data. Match tolerances of MS1 m/z and RT matches were set as 15 ppm and 30%, respectively. MS2 spectra of qualified peaks were further matched against the surrogated MS2 spectra from seed metabolites (constraint 4). The qualified peaks with dot product score larger than 0.5 or matched fragments more than 4 were linked to the seed metabolites, and their putative structures were assigned from the reaction-paired neighbor metabolites.”*

Comment #6: *“P7 L155-157 "global peak correlation network"  following previous remarks, now the authors state that their implementation of feature grouping (i.e., global peak correlation network (network 3), akin to CAMERA and IIN) is unique. However, they do not explain or highlight in the main text how it is unique. To this reviewer, not everything needs to be novel/unique to make an approach useful; however, when this is claimed, it has to be explained. Please elaborate and clearly explain what is unique. The authors also state here that it "effectively optimizes putative metabolite annotations" - did the authors compare their approach to, e.g., CAMERA? Is it much better? Again, this is not a necessity for this reviewer, but it is good practice to build on (and implement) good functioning packages and only build new ones when it is needed. Also, when it becomes clearer which aspects improved the results, complementary tools could also adopt them.”*

Ans: Thanks a lot for the reviewer’s comment. Global peak correlation network in KGMN (referred as KGMN) and CAMERA have different objectives and designs although they both could be used for annotating the ion forms of features (i.e., abiotic peaks in previous version). Specifically, our global peak correlation network aims to comprehensively recognize different ion forms for putative annotated metabolites in the network 2, and further improve the accuracy of metabolite annotation. In brief, our approach used the putative annotated peaks as base peaks, then targeted reorganized the different ion forms of peaks. These ion forms composed a subnetwork, which finally facilitated to optimize and filter unreliable metabolite annotations in the network 2. In another word, in our global peak correlation network, we employed a targeted approach to annotated different ion forms of features (aim 1) and improved the metabolite annotation accuracy in the whole workflow (aim 2). Different with KGMN, CAMERA is only designed to annotate different ion forms globally in the peak table. Starting from the most intense peak,

CAMERA groups all peaks via RT grouping and a graph-based algorithm, then annotates different ion forms. It is an untargeted approach to automatically annotate different ion forms and aims to reduce the data complexity.

To address this reviewer’s comment, we used the benchmark data sets (the dataset used in Figure 2) to compare the performance differences between CAMERA and KGMN. Only annotations of ion forms were used for evaluation because CAMERA does not have the capability for metabolite structural annotation. As shown in **Supplementary Fig. 7a**, KGMN annotated 3,374 out of 3,451 peaks (97.8%), and 3,325 out of 3,374 peaks (97.5%) had correct annotations in ion forms. As a comparison, CAMERA annotated 2,297 out of 3,451 peaks (66.6%), and 1,877 out of 2,297 peaks (81.7%) had correct annotations in ion forms. We think the higher coverages and correct rates of KGMN benefit from its targeted recognition approach. In particular, KGMN has excellent performances in recognizing inexplicable in-source fragmentation ions and neutral losses (**Supplementary Fig. 7b**).

We have added related descriptions in the revised manuscript as below: “We also compared the global peak correlation network in KGMN with CAMER⁴⁴. Comparing to CAMERA, KGMN annotated more ion forms (3,374 vs 2,297, KGMN vs CAMERA), and have higher correct rate (97.5% vs 81.7%, KGMN vs CAMERA) (**Supplementary Fig. 7**). In particular, KGMN has excellent performances in recognizing inexplicable in-source fragmentation ions and neutral losses (**Supplementary Fig. 7b**).”

Supplementary Figure 7. Benchmark comparison between CAMERA and KGMN for annotating ion forms of metabolic peaks. (a) Percentages of annotation coverage and correct/error rates for annotating ion forms of metabolic peaks. (b) Annotation percentages for different types of ion forms. The R package “CAMERA” (v1.46.0) and the same rule table were used for evaluation.

Comment #7: *"P17 L401-402 "Hooft group"  based on the cited reference, it is likely that the authors mean the "Van der Hooft Computational Metabolomics Group". If so, please adjust."*

Ans: Thanks a lot for the reviewer's correction. We have modified the "Hooft group" as "Van der Hooft Computational Metabolomics Group" in the revised manuscript.

Comment #8: *"GitHub repository - please note that on the main page, it is noted that the sample csv file needs "three" columns, whereas it should be "two" (according to the tutorials, and to the two column names that are specified)."*

Ans: Thanks a lot for the reviewer's careful correction. We have corrected the typo in the README file in GitHub repository.

REVIEWERS' COMMENTS

Reviewer #3 (Remarks to the Author):

Thanks for addressing the additional set of items raised by all the reviewers. I realise it is not easy to explain the workings of such a complex tool within the imposed format, but I think you did a good job, especially after this round. I do hope you have found my remarks and concerns constructive. Again, it is great to see such computational metabolomics tools being developed.

I do agree with the authors that benchmarking metabolomics tools remains challenging with the various datasets (and their formats and the instrument types) and different mass spectral libraries that are used. Nevertheless, the NetID and CAMERA comparisons provide the reader a handle on the benefits of KGMN and the suitability on their own metabolomics datasets. Thus, thanks for making these additions to the new version.

Also, the scope of your work (and its limitations) are now better addressed in the abstract and discussion. Please find a title correction and suggestion below.

Here are some final suggestions to further increase the clarity and impact of your work:

- Title of the manuscript: at least at "ing" after metabolic network (currently grammatically incorrect)
 Metabolite annotation from knowns to unknowns through knowledge-guided multi-layer metabolic networking

and to make the scope of your work clearer, as a response to the authors' request for ideas how to indicate this, here is my (strong!) suggestion (i.e., adding "in common biological samples" - most metabolomics researchers will directly understand what that means!) 

Metabolite annotation from knowns to unknowns through knowledge-guided multi-layer metabolic networking in common biological samples

- Include in the Code Availability Section how (some of) the figures can be made; i.e., if I am not mistaken, Figure 4b (which I think is very elegant) can be created with code from <https://github.com/ZhuMetLab/MetDNA2InSilicoTool> - and the same is probably true for other

figure(s) (panels) (also using <https://github.com/ZhuMetLab/MetDNA2Vis>). Some guidance here would be appreciated by the community.

- Apply a final "English polish" as there are still some sentences and phrases that could be streamlined (further) by removing grammatical inconsistencies - i.e., improving the general readability may further enhance the impact of your work. See for example the title correction above, and at least carefully check the abstract, discussions, and conclusions.

Kind regards from Wageningen, the Netherlands,

Dr Justin van der Hooft

Response to the reviewers:

The authors would like to thank the reviewer for the helpful comments. We feel these comments have strengthened the manuscript considerably.

Reviewer #3:

Remarks to the Author: *“Thanks for addressing the additional set of items raised by all the reviewers. I realise it is not easy to explain the workings of such a complex tool within the imposed format, but I think you did a good job, especially after this round. I do hope you have found my remarks and concerns constructive. Again, it is great to see such computational metabolomics tools being developed.”*

Ans: We appreciate the reviewer’s positive comments toward publication. We feel these comments have significantly strengthened the manuscript.

Comment #1: *“Title of the manuscript: at least at “ing” after metabolic network (currently grammatically incorrect)  Metabolite annotation from knowns to unknowns through knowledge-guided multi-layer metabolic networking,*

and to make the scope of your work clearer, as a response to the authors’ request for ideas how to indicate this, here is my (strong!) suggestion (i.e., adding “in common biological samples” - most metabolomics researchers will directly understand what that means!)  Metabolite annotation from knowns to unknowns through knowledge-guided multi-layer metabolic networking in common biological samples”

Ans: Thanks a lot for the reviewer’s suggestion. The title of the manuscript has been modified to *“Metabolite annotation from knowns to unknowns through knowledge-guided multi-layer metabolic networking”*.

Comment #2: *“Include in the Code Availability Section how (some of) the figures can be made; i.e., if I am not mistaken, Figure 4b (which I think is very elegant) can be created with code from <https://github.com/ZhuMetLab/MetDNA2InSilicoTool> - and the same is probably true for other figure(s) (panels) (also using <https://github.com/ZhuMetLab/MetDNA2Vis>). Some guidance here would be appreciated by the community.”*

Ans: Thanks a lot for the reviewer’s suggestion. We have added the guidance of these packages in the Code Availability Section as below: *“The source codes of in-silico MS/MS validations (MetDNA2InSilicoTool) were provided in GitHub [<https://github.com/ZhuMetLab/MetDNA2InSilicoTool>] and Zenodo [<https://doi.org/10.5281/zenodo.7233184>]⁷⁵. The source codes of multi-layer network*

visualization (MetDNA2Vis) were provided in GitHub [<https://github.com/ZhuMetLab/MetDNA2Vis>] and Zenodo [<https://doi.org/10.5281/zenodo.7233189>]⁶.”

Comment #3: *“Apply a final “English polish” as there are still some sentences and phrases that could be streamlined (further) by removing grammatical inconsistencies - i.e., improving the general readability may further enhance the impact of your work. See for example the title correction above, and at least carefully check the abstract, discussions, and conclusions.”*

Ans: Thanks a lot for the reviewer’s comment. We have carefully reviewed and modified the sentences and phrases in the revised manuscript.